

# Reduction Assessment of Agricultural Non-Point Source Pollutant
# Loading
YiCheng FU [1]*, Wenbin ZANG[1], Jian ZHANG[1], Hongtao WANG[2], Chunling ZhANG[1] and
Wanli SHI[1]
1. State Key Laboratory of Simulation and Regulation of River Basin Water Cycle, China Institute of Water
Resources and Hydropower Research, Beijing, 100038, China
2. Yellow River Conservancy Technical Institute, Kaifeng Henan, 475004, P.R. China
* Corresponding author, E-mail: swfyc@126.com
**Abstract:**
NPS (Non-point source) pollution has become a key impact element to watershed environment at present.
With the development of technology, application of models to control NPS pollution has become a very
common practice for resource management and Pollutant reduction control in the watershed scale of China.
The SWAT (Soil and Water Assessment Tool) model is a semi-conceptual model, which was put forward to
estimate pollutant production & the influences on water quantity-quality under different land development
patterns in complex watersheds. Based on the overview of published papers with application of SWAT, the
study topics is mainly focus on nutrients, sediments, impoundment & wetlands, hydrologic characteristics,
climate change impact, and land-use change impacts. SWAT model was constructed based on rainfall runoff
and land use type. The migration-transformation processes of agricultural NPS pollution was simulated and
calculated based on the SWAT model. Besides, the loadings and distribution traits of NPS pollutants were
also systematically analyzed based on the model. The model was used to quantify the spatial loading
intensities of NPS nutrient TN (Total Nitrogen) and TP (Total Phosphorus) to HTRW (Huntai River
Watershed) under two scenarios (without & with buffer zones). The SWAT model was validated using actual
monitoring information as well as the physical properties of the underlying substrate, hydrology,
meteorology and pollutant sources in the HTRW. Scenario settings are mainly based on the changes of
surface runoff and sediments, climate and land-use change from different spatial scales, and climatic/



physiographic zones. About 1 km within both banks of the trunk streams of the Huntai, Taizi and Daliao
rivers, and 5 km surrounding the reservoirs were defined as buffer zones. Existing land use type within the
buffer zone was changed to reflect the natural environment. The output of pollutant production under the
EPS (Environmental Protection Scenarios) was calculated based on the status quo scenario. Under the status
quo scenario, the annual mean modulus of soil erosion in the HTRW was 811 kg/ha, and the output
intensities of TN & TP were 19 & 7 kg/ha, respectively. For the unit area, the maximal loading intensities
for TN & TP were 365.36 & 259.83 kg/ha, respectively. In terms of spatial distribution, TN & TP loading
varied substantially. Under the EPS, the magnitude of N & P production from arable land decreased to a
certain degree, and the TN & TP pollution loading per unit area were reduced by 5 & 1 kg/ha annually,
respectively. In comparison, the quantity of NPS pollutant production under the EPS was reduced by 21.9%
compared with the status quo scenario, and the quantities of TP & TN decreased by 10.4% & 25.9%,
respectively. These changes suggested a clear reduction in the export loading of agricultural NPS pollution.
Loading intensities analysis showed that land use type is one key factor for controlling NPS pollution. The
NPS pollutant loading decreased under the EPS, which showed that environmental protection measure
could effectively cut down NPS pollutant loading in HTRW. SWAT was used to assess the reduction of
agricultural NPS pollutant. However, SWAT model requires a large amount of data about the watershed
being modeled; the data inaccuracy and local factors would impact the accuracy of the SWAT model. To
determine the pollutant reduction under different land development patterns, and examine uncertainty of
sensitivity parameters, SWAT model in China has wide range of potential application.
**Key words**:
agricultural NPS pollutant loading; Huntai River Watershed; status quo scenario; environmental
protection scenarios

# 1. Introduction

NPS pollution has become key influencing factor to improve surface water quality. There are many
literatures have illustrated that underlying surface condition & precipitation characteristics will impact the





spatial distribution characteristics of NPS pollution nutrient loading (Robinson et al.,2005; Lindenschmidt
et al.,2007). The pollutant production from different land use types vary substantially (Niraula et al.,2013).
The concentrate on NPS pollution is dependent on discharge it is highly variable and does not enable a fair
comparison between different areas (Tucci 1998; Dingman 2002; de Oliveira et al.,2016). Loadings are
considered better for comparing watersheds and for establishing the relationship between pollutants and
land use (Quilb´e et al.,2006). At present, many researchers have preferred loadings over concentrations to
convey their research (Yang et al., 2007; Ouyan et al.,2010; Outram et al., 2016). Land use types &
underlying surface condition will influence the resources and nutrients distribution, and which will result
in the reduction of NPS pollutant loading (Hundecha et al.,2004; Ahearn et al., 2005; Ouyang et al., 2013).
In general, the spatial-temporal characteristic of NPS pollutant can be studied based on data statistics &
model simulation (Shen et al.,2013a). SWAT model can be determined NPS pollutant loading & supplied
the decision-making program for watershed comprehensive development (Shen et al.,2011). Many
documents have confirmed the combination of different land development patterns & landscape
characteristics could reduce NPS pollution (Seppelt et al., 2002; Sadeghi et al.,2009).
Distributed physics & semi-conceptual models are effective means to calculate and assess the NPS
pollution spatial loading intensities. At the end of the $20^{th}$ century, the SWAT model was developed by
American scientists of USDA-ARS (Arnold et al.,1998). SWAT has been widely used in runoff simulation,
the calculation of NPS pollution & implementation of best management practices. The SWAT was widely
used in assessing the impact of NPS pollution under different land use types, for which was consisted by
underlying surface, vegetation coverage, hydrometeorology, and agricultural production modules. The
production changes of agricultural NPS nutrients based on diverse land development patterns have been
studied & analyzed by SWAT model (Ficklin et al.,2009; Shen et al., 2013b; Geng et al., 2015). The main
body of SWAT model includes 701mathematical equations & 1013 intermediate variables, which has been
widely used to calculate & assess the distribution traits of NPS pollutant loading, as well as analyze the
effects of land use and its spatiotemporal distribution pattern on NPS pollutant & soil loss in watershed
scale (Mapfumo et al.,2004; Gosain et al.,2005; Ouyang et al., 2009; Logsdon et al.,2013).



The HTRW is the important tributary of Liaohe River Basin, which has been polluted seriously in recent years. The main NPS pollution in Liaohe River is agricultural NPS pollution, and most NPS pollution happens in HTRW within Liaoning province (Liaoning Province DEP, 2011). Therefore, the HTRW face immense pressure due to water pollution. According to the twelfth five-year developmental plan, the annual mean growth of GDP in the Liaohe River watershed was greater than 13% and the urbanization rate was close to 75%. The policy of 'Revitalization of Old Industrial Bases in Northeast China' has caused significant changes in the land-use structure (Liu et al.,2014). This accelerating urbanization alters the existing land use type in a way that results in more NPS pollution to local surface waters (Kuai et al.,2015). HTRW is the Basic product manufacturing base in China.

The SWAT of the present study was used to quantify the spatial loading intensities of TN & TP to HTRW under different land use types, and assess the adaptability changes based on NPS pollutant loading reduction. Nutrient losses were simulated in different scenarios-status quo scenario (without buffer zones) and "environmental protection" scenario (EPS, with buffer zones), using SWAT. The flow chart of this study was to: (1) elaborate the underlying surface (land use) changes in the HTRW; (2) simulate the NPS pollution loading (TP & TN) of the HTRW under two scenarios; (3) contrast the different of NPS pollution loading in two scenarios, and assess the effect of reducing pollution loading under EPS. In this paper, the SWAT was used to estimate the agricultural NPS pollution loading of HTRW, and digital comparison analysis method was utilized to analyze the spatial distribution characteristics of pollution loading.

## 2. Materials & methods

### 2.1 HTRW

The HTRW (40°27′~42°19′N, 121°57′~125°20′E) is in Liaoning province (Northeast China), and the watershed area is $2.73 \times 10^4$ km², which takes about 1/5 of the area of Liaoning province (Fig 1). The HTRW is a tributary of Liaohe River Basin (The Liaohe River Basin is one of China's larger water systems) and is consist of Hunhe River, Taizi River, and Daliao River. The Hunhe River, Taizi River, and Daliao River watershed is HTRW's sub-watershed. The HTRW has varied topography, low mountain is located in eastern part, and the other parts are



alluvial plain. The elevation of northeast region is high. Loamy soils are mainly distributed in alluvial plain, and

the average grade of lower HTRW is about 7%. HTRW area includes the cities of Fushun, Shenyang, Benxi,

Liaoyang, Anshan, and Yingkou, most of Panjin city, some portions of Tieling city and a minor portion of

Dandong city. The maxim runoff in the watershed is $76.32 \times 10^8$ m$^3$, primarily concentrated in June through

September. The stream flow and nutrient were validated based on the five monitoring stations, Beikouqian,

Dongling Bridge and Xingjiawopeng are located in Hunhe River, Xialinzi and Tangmazhai are in Taizi Rive. The

total population of HTRW is 18.9 million people. The GDP is about 62% of Liaoning Province in 2012. HTRW

has temperate continental climate, the average annual temperature is 7°C, and precipitation is 748 mm.

The HTRW is in a conventional agricultural farming area, with a large area of farmland dominated by

crop plants. The total area of farmland is 10 763 km$^2$ (account for 39.4% of the total area), including

4 086 km$^2$ of paddy field (dominated by rice) and 6 677 km$^2$ of dry farmland (including corn, soybean,

vegetables and other crop plants). The upper reaches of the Hunhe and Taizi rivers have mountainous (69%),

low hilly (6.1%) and plain land (24.9%). The economic output value of HTRW is dominated by agriculture.

The farmland is mainly distributed in the floodplain area and valleys in riverine belts. Considering land

pattern, rainfall and source of pollutants, the HTRW faces a high risk of pollution from agriculture. Heavy

use of fertilizers and soil erosion in the upper of HTRW has led to serious NPS pollution in HTRW. For

example, the Dahuofang reservoir of the Hunhe River and the water resources conservation area in its upper

reaches are facing multiple threats, the agricultural NPS pollution is becoming increasingly serious and has

not yet been controlled effectively (Shen et al., 2013c).

Fertilization in the HTRW is predominantly with nitrogen, followed by phosphorous and potassium. The

heavy use of chemical fertilizers was mainly urea, diammonium phosphate and a small amount of potassium

phosphate compound fertilizer. Atrazine and acetochlor were mainly used on dry farmland, and butachlor

was mainly used in paddy fields. Based on the statistical data for 2006-2012, the quantity of fertilizers and

pesticides applied in the watershed fluctuated annually. The upper reaches of the Huntai and Taizi rivers

are dominated by mountains, the cultivation and harvesting of crops are conducted by hand, and therefore

thorough statistics are not available. At present, weeds and pests in farmlands were mainly controlled by



pesticides and herbicides. The upstream is rich in forest resources, the downstream has a large number of
farmland, special landscape layout makes the HTRW become potential area for agricultural NPS pollution.

## 2.2 Model description

### 2.2.1. SWAT principle

SWAT is a semi-physical model developed to quantitatively calculate the response status of water
quantity & quality to land use and management methods in the scale of watershed (Gassman et al.,2007).
SWAT is an effective to determine the long-term impact using monitoring data (Arnold et al.,2012). The
basic data input for model running includes DEM (Digital Elevation Model)/topography, soil type,
vegetation status/Land landscape, and BMPs (Best Management Practices scenarios). The calculation unit
of watershed SWAT model is sub-watershed, and HRU (Hydrological Response Units), the unit delineation
is based on the underlying surface status, vegetation coverage, soil classification, and land use (Neitsch,

139    2005).

The HRUs of SWAT are automatically divided according to soil conditions, DEM, geomorphological
features, and land development (Douglas-Mankin et al., 2010). For the calculation process is realized on
HRU, therefore, we selected 0% land development, elevation/slope, and soil classification/attributes as the
initial value in the scale of HTRW, therefore, 184 HRUs were delineated to determine NPS pollutant
loading. In order to assess pollutant loss and ecological flow status, the flow curve, soil nutrient loss curve,
and water-salt balance equation were applied during the period of model debugging. Meteorology data (sun
radiation, atmospheric pressure, atmospheric temperature, precipitation and wind speed) were obtained
from meteorological and hydrological stations of 12 cities located within HTRW. The data of BMPs, such
as crop sowing/harvest time, crop irrigation time, cultivation structure of cultivated land, fertilizer-use
efficiency, and farmland planting plan were got from agriculture & environmental management department,
or collected from the survey of farmers status quo. Based on the above assessment results, we used
QUAL2E (water quality model) to determine N & P yields loading, the route of sediment transport, and
pollutant concentration of watershed outlet.



The SWAT is mainly used to assess the N & P production, migration, and transform. These cycling
processes occur simultaneously with the processes of the hydrological cycle and soil erosion. The N & P
cycles simulation of SWAT was developed based on 5 different forms of N and 6 different forms of P,
respectively. The N & P cycles were consisted of the process of decomposition, mineralization, fixation,
and conversion. The NPS pollutant loading function is the basis of assessing N & P transport and
transformation (McElroy, 1976; Williams et al.,1978; Zhang, 2005). Organic N & P losses calculation of
SWAT was achieved by the integrated function of soil nutrient curve, NPS pollutant loading, soil properties
change rate, and crop growth characteristics. The total amount of nitrate in lost soil was calculated by the
product of water volume and nitrate concentration in water. Water volume is the consisted of surface runoff,
groundwater runoff, and interflow/subsurface flow. The soluble P removed in runoff is estimated using the
P concentration in the top soil layer, runoff volume and the P soil partitioning coefficient. The concentration
of soluble P in water is calculated by topsoil P stocks, runoff variation, ratio of soluble P, and soil particle
characteristics.
Surface runoff from daily precipitation in HRU/Sub-watershed was calculated & assessed using the SCS-
CN corresponding relationship curve and rainfall-runoff Coefficient (USDA Soil Conservation Service.
National Engineering Handbook, 1972). With SCS-CN curve, saturated moisture, soil water profile/vertical
distribution of soil moisture content, runoff module number of the underground water is determined, as
well as the related parameters daily of precipitation. The total discharge of runoff from sub-watershed/
HRUs is the sum of surface runoff flow, groundwater runoff flow, and interflow/subsurface flow. Domestic
water & irrigation water is direct consumptive water resources, the mainly water resources is surface runoff
& groundwater runoff (Neitsch,2005). The main routing of water circulation in SWAT is network-node
diagram and natural-artificial dualistic water cycle mode. In the paper, we used a dualistic method for multi-
layer and multi-function separation and interception of the rainfall and run off resources. Circulating flow
of SWAT was varied with the dynamic changes of evaporation, infiltration, transport, and return flow
(Arnold et al.,1998). The HRUs of SWAT used soil erosion modulus, soil & water loss coefficient, and
Universal Soil Loss Equation (MUSLE) to analyze erosion and sediment yield (Williams, 1975). Sediment





is routed through channels using Bagnold's sediment transport equation (Bagnold, 1977). We used 2009
version of SWAT to calculate the correlation parameters.

## 181 2.2.2. Model data input

The data of DEM, geomorphology, underlying surface status, soil properties, land cover, meteorological
& hydrological data (precipitation, evaporation, temperature, and atmospheric pressure, et al.) were input
to achieve the operation of SWAT (Niraula et al.,2013). Figure 2 supplied the basic data information to be
used in SWAT model. We used $30 \times 30$ grid data (elevation) as the basis for DEM operation. The DEM
was selected as the topographical basis on which to construct the SWAT model, to extract the scope of the
study area and to construct the topographical model. The stream network in the study area was extracted
using 1:250 000 digital water system data (data source: www.geodata.com) as an ancillary model to
construct the stream network model of the HTRW. We classified land use types into 27 categories. The
main type of land use of HTRW is forest (including orchard, 48.64%), dry land (24.38%), rice paddy
(14.92%), urban land (vacant land, 7.78%) and unused land (uncultivated land, 1.85%) grassland (0.92%).
Soil types were categorized into 26 types, the primary soil types in this area are brown soil (54.1%), meadow
soil (29.7%) and paddy soil (11.0%). The database of the underlying substrate was constructed based on
the database of soil types using the soil properties & land development data as underlying substrate
parameters (Liu et al.,2015). The soil parameters were obtained from National earth system science data
sharing infrastructure database. The watershed meteorological data (precipitation, evaporation, and
temperature) used in the present study include precipitation data for 1990-2009 collected by 76 rainfall
stations and air temperature data for1990-2009 collected by 12 city meteorological stations. We used
meteorological monitoring data for the simulation of precipitation & evaporation. The missing
meteorological information (rainfall, humidity & atmospheric pressure, air temperature, solar radiation &
wind speed data) can be generated using the weather data generator simulation. At least 3 sets monthly
monitoring data of nitrate ($NO_3$), nitrite ($NO_2$), Ammonia ($NH_3$, $NH_4$), TP, and TP, were available in the
time of 2006–2009. We got the information of crop type, farming system, sowing time, fertilization time,
and social economics from investigation and statistics department in HTRW. All the data were validated





by the standard procedures used by the SWAT.
The data information (type, scale, description, and source) of SWAT in HTRW are showed in Figure 2.
We input the related meteorological and soil data of SWAT got from China Meteorological Administration
and Environmental-Ecological Science Data Center for West China. The China Hydrology, water resources
& water quality monitoring department of HTRW provided the automatic & regular monitoring
hydrological data. The Liaoning province Water Resources Administrative Bureau granted permission for
the modelling of the pollutant production response to different land utilization scenarios in the HTRW.

### 2.2.3. Calibration and validation

The data of monthly scale were used to achieve the simulation of SWAT. We used the code open SWAT-
CUP module to calibrate parameters of SWAT in HTRW automatically (Abbaspour et al.,2007). Sequential
uncertainty fitting algorithm has higher calculation accuracy and simple application method, which was
extensive used in the SWAT-CUP module (Wang et al.,2014; Yang et al.,2008). The $E_{NS}$ can effectively
avoid the uncertainty of hydrological sequence (precipitation, water flow, and evaporation), which was
used to evaluate the run-off flow change of hydrological station in HTRW (Nash, 1970).
The model for the present study was calibrated and tested using artificial parameter modification and
automatic calibration. First, the runoff was calibrated, followed by N, P and other nutrients. The runoff was
calibrated and tested using real data from the Xingjiawopeng, and Tangmazai hydrological station (Figure
4). The simulated values of N and P were calibrated using monitoring data from Beikouqian, Dongling
bridge, Xingjiawopeng, Xiaolinzi, and Tangmazhai hydrological station. Various hydrologic and water
quality parameters were adjusted under their change interval to fit with the monitored/observed data during
calibration and validation (Figure 3). ESCO, GWQMN, and SURLAG were three key parameters in the
process of calibration & validation of water flow (Shen et al., 2010; Francos et al., 2003). The other sensitive
parameters selected for calibration & validation in HTRW were showed in Figure 3. In the HTRW,
Liaoning Province government began to monthly monitoring of pollutant since 2006. The runoff, TN & TP
loadings data used for calibration & validation were from 1992 to 2009, from 2006 to 2008, respectively.
In the present study, the simulated effects were evaluated based on analysis and comparison using the



runoff hydrograph, $D_v$ (relative deviation), $E_{NS}$ and $R^2$ (certainty coefficient). The runoff hydrograph and
$D_v$ were frequently used to simulate the entire deviation of water quantity; $E_{NS}$ and $R^2$ were used to simulate
the effects of the simulation (Yang et al.,2014). The $D_v$, $E_{NS}$ and $R^2$ are calculated as
$$D_v = [(M - W)/W] \times 100\% \tag{1}$$

Here, $D_v$ was the relative deviation; $W$ was the observed mean value; and $M$ was the predicted mean value.
$$E_{NS} = 1 - [\sum_{i=1}^{n}(W_i - M_i)^2 / \sum_{i=1}^{n}(W_i - \overline{W})^2] \tag{2}$$

Here, $E_{NS}$ was the Nash-Sutcliffe efficiency coefficient; $W_i$ was the observed data at $i^{th}$ period; $M_i$ was the
simulated data at $i^{th}$ period; and $\overline{W}$ was the observed mean value.
$$R^2 = \{[\sum_{i=1}^{n}(W_i - \overline{W})(M_i - \overline{M})]/[\sqrt{\sum_{i=1}^{n}(W_i - \overline{W})^2}\sqrt{\sum_{i=1}^{n}(M_i - \overline{M})^2}]\}^2 \tag{3}$$

Here, $R^2$ was the certainty coefficient; $W_i$ was the observed value at time $i$; $M_i$ was the simulated value at
time $i$; $\overline{W}$ was the observed mean value, and $\overline{M}$ was the predicted mean value.
The first four years (1990-1994) were regarded as domestication stage of SWAT to minimize the
uncertainty of initial meteorology & underlying surface value. We used manual method of parameter
adjustment to calibrate the SWAT in HTRW. To determine the sensitivity of various parameters, we
manually adjusted one parameter at a time according to the accuracy and change interval in Figure 3. To
realize the matching between hydrographs base flow from model simulation and actual monitoring, the
quantitative data analysis technology ($E_{NS}$ & $R^2$) was used to calibrate SWAT. In order to calibrate the
stream flow, we subsequently calibrated runoff, and nutrients (TP & TN) with the same geographical and
hydrological data. During calibration, we used LOADEST model to eliminate the uncertainties caused by
the differences in sampling & testing methods of water quality (Yang et al.,2014).
## 2.3 Scenarios setting
To seek the relationship between agricultural NPS pollutant loading and land use types, comprehensive
comparison method was used in different land use types under urbanization. In this study, two scenarios were
established: status quo scenario, and "environmental protection" scenarios (EPS).



The status quo scenario was formulated based on the existing socio-economy developmental structure
and environmental protection measures, and the land use type in the light of the existing developmental
model and planning conditions. The BMPs information & land use data (cultivated land area, pesticide &
fertilizer use utility amount, crop type) were obtained from Liaoning Province statistical yearbooks-2013
and field survey.
Considering the regional development prospects & eco-environment protection strategy in HTRW, the
EPS was proposed. 1 km within both banks of the Hunhe, Taizi and Daliao rivers and 5 km surrounding
reservoirs are defined as buffer zones. In the buffer zones, existing land use types were changed to restore
the natural environment (grassland and forest). The output of pollutant production is calculated based on
the regional environmental protection. This scenario not only preserves the fundamental position of
agriculture in the watershed, but also improve the ecosystem service value of the watershed by only slightly
reducing the amount of fertilizers and pesticides used for agricultural production. The scenarios setting can
provide scientific basis for further understanding characteristics of the nitrogen and phosphorus loadings
and agricultural structure adjustment in HTRW.

## 269 2.4 Study framework

Hunhe River, Taizi River, and Daliao River sub catchment was delineated based on DEM & river system,
and further divided by 29 small calculation modules according to 184 HRUs, water resources zoning, and
administrative zoning. According to the water network & the location of basin drainage, we used the monitored
data calibrate & validate the stream flow and concentration changes of pollutants in HTRW. And then the land
development patterns in two scenarios were imported to SWAT model to simulate the TN and TP pollution
loading. Finally, the NPS pollution loading decrease was analyzed based on land use scenarios.
The primary source area of aquatic pollution is mainly distributed along both channels of the trunk stream
of the Hunhe River, Taizi River, and Daliao River; the risk of NPS pollution is mainly related to the patterns
of agricultural plantation and farmland utilization. The secondary source area of aquatic pollution is mainly
distributed along the tributaries of HTRW. Therefore, this project paid special attention to the pollutant



production in the agricultural lands adjacent to the water channels.

# 3. Results & discussions

## 3.1 Modelling validation

**Stream flow.**    Because of HTRW lacks basic runoff data, the present study focused on calibrating and
testing the runoff model. During annual calibration, the runoff curve data were first calibrated, and then the
available water content in the soil and the soil evaporation compensation coefficient were modified until
they matched the requirements for runoff. Finally, the monthly runoff curve was modified. For the
simulation, 1990-1994 was the model preparation period, 1995-2001 was the model calibration period, and
2002-2009 was the model validation period.
According to the calibration results, $E_{NS}$ and $R^2$ for Xingjiawopeng hydrological station and Tangmazhai
hydrological station were both greater than 0.6, and the $|Dv|$ values for both stations were less than 20%
during the model preparation period, suggesting that the parameters of the SWAT model were reliable after
calibration, and thus the model can be used for further study. The monitoring value fitted very well with
the simulation value obtained from hydrographic curve, most crest values observed were very similar. In
the model calibration period, the matching curves for the simulated and measured values of monthly runoff
at Xingjiawopeng and Tangmazhai hydrological stations are shown in Figure 4 (a) and Figure 4 (b). The
runoffs at these two hydrological stations were well matched. However, the accuracy of the runoff in the
second half of the year in 2002, 2005 and 2006 was poor, likely due to the length of the data series and
specific stations selected. In terms of the standards for the simulation and evaluation of the hydrological
model, the simulation effects at the monthly scale were much better.
**Nutrients.** The nutrients concentrations of water were simulated by SWAT. Based on the verification of
the accuracy of the initial concentrations, the fertilization and cultivation measures for nutrients in the soil,
the nitrate and soluble P loading can be simulated by adjusting the nitrogen permeability coefficient
(NPERCO) and the phosphorous permeability coefficient (Lam et al., 2011). Beikouqian, Xingjiawopeng,
Xiaolinzi and Tangmazhai four hydrological stations had a continuous monthly water quality monitoring



data from 2006 to 2007. Only the monthly data of TN & TP in Beikouqian were validated from 2008 to
2009 for the insufficient of water quality monitoring data. The Xingjiawopeng, Xiaolinzi and Tangmazhai
Hydrological Stations only had the TN data in the study time. Therefore, Beikouqian was selected to show
validation curves, the TN $E_{NS}$ and $R^2$ were 0.64 and 0.78, and the TP $E_{NS}$ and $R^2$ were 0.60 and 0.75,
respectively (Figure 5(a), Figure 5(b)). The calculation results of $E_{NS}$ and $R^2$ of Xingjiawopeng, Xiaolinzi
and Tangmazhai hydrological stations were 0.62 and 0.73, 0.61 and 0.72, as well as 0.62 and 0.77,
respectively. The values of all $R^2$ were higher than 0.7, which confirmed the SWAT could be used for water
quality simulation in HTRW.

## 3.2 NPS pollutant loading under status quo scenario

The output of NPS pollutant production was calculated using the pollutant loading approach based
on the attributes of the regional calculation results and land use scenarios in HTRW. The output of N &
P production in different calculation units were calculated based on the spatial changes of soil types, crops
and residuals, as well as the differences in the coefficients of N & P losses under different land use types.
The paddy fields, rural residential, urban development, and vegetation type maybe the important indicators
for variability in NPS pollution, and that nutrition pollution was influenced by the integrated effects of
different land uses (Cai et al., 2015; Lee et al.,2010). The annual throughputs of TN & TP production
were 18 707 t and53 322 t, respectively (Table 1).

### 3.2.1. Sediment

The sediment loading is the data basis to calculate the TN & TP loading, and which is affected by the
type of land development and vegetation coverage (which was generally dominated by forest and farmland).
Based on the simulation by the SWAT model, the annual output of sediment (silt) production in the
watersheds of the Hunhe, Taizi and Daliao rivers was $22\times10^4$ t, $170\times10^4$ t and $30\times10^4$ t, respectively. The
annual soil erosion modulus in the study area was 0.811 t/ha, and its spatial distribution is shown in Figure
6(a). The soil erosion (sediment) value varied widely in different regions, with the change interval from 0
to 1.824 t/ha. Soil erosion in Daliao River watershed was very serious (with up to 1.568 t/ha in some



regions), followed by the Taizi River watershed (The amount was 1.223 t/ha in most regions) and Hunhe
River watershed (Less than 0.19 t/ha in most regions). Yingkou and Dashiqiao has even topography, and
incoming silt from the upper reaches is accumulated therein. The soil erosion modulus is therefore very
high, which contributes greatly to the silt inputs to the HTRW (Tang et al.,2012). The soil erosion was
affected by natural & human factors. The natural factors mainly included topography, underlying surface
conditions and soil types, the human factors mainly consisted of vegetation coverage, precipitation type,
land use, crop cultivation and cultivated land farming methods. Moreover, mountainous area has great soil
erosion (Hong et al.,2012). The HTRW had high forest coverage, which effectively prevented the soil
erosion. Daliao rivers had a large area of cultivated land, therefore, there was higher probability to cause
soil erosion. Besides, the soil types are also the key influencing factors to cause soil erosion, therefore, the
brown and paddy soils are prone to bring about the accumulation of sediment (Hong et al.,2012).

### 3.2.2. TP

With SWAT simulation results, the annual output of TP production in the watersheds of the
Hunhe, Taizi and Daliao rivers was 8 993 t, 6 399 t and 3 315 t, respectively, the watershed
loading output intensity was 7 kg/ha. The TP loading had the same spatial distribution pattern
with the sediment loading. The TP loading ranged from 0 to 259.83 kg/ha. Figure 6(b) showed
the spatial variation of TP loading the HTRW. The average annual water volume was affluent
in Hunhe River, which prompted a large amount of P deposited in the downstream plain. The
changes in space of the TP loading was affected by topography, precipitation, land use type,
and silt losses. The TP loading output intensity of on the slope in the Daliao River watershed
was higher than that in the Hunhe River watershed, and the Taizi River watershed was the
lowest. Large amounts of fertilizer and pesticides have been applied to the farmland.
Organophosphate pesticides accounted for 40% of the total pesticides. Therefore, the farmland
has high TP concentrations, which was the same findings with Wang (2012). The paddy fields,
brown soil and dry lands mainly distributed in Hunhe River downstream, therefore, the P
loading in these plain area is higher (Li et al., 2010). Correspondingly, the cities and counties



with a large proportion of farmland, such as Dashiqiao, Panshan and Dawa city in the Daliao
River watershed, as well as the city of Haicheng and Taian county in the Hunhe River watershed,
have higher TP loading output intensity. The regions with a large proportion of developed land,
such as the city center of Fushun, Shenyang in Hunhe River watershed, the municipal districts
of Liaoyang city and Benxi city at the Taizi River watershed, which have lower TP loading
output intensities. Based on the land use type, the tributaries with a higher proportion of
farmland have the highest TP output intensities, whereas the tributaries with substantial
vegetation cover as forested land have relatively lower TP output intensities. The output
intensity of TP is closely related to soil characteristics and attributes.

### 3.2.3. TN

Upon simulation and calculation, the output of TN production in the watersheds of the Hunhe,
Taizi and Daliao rivers was 24 264 t, 19 010 t and 10 048 t. The annual loading output intensity
of TN in the watershed was 19 kg/ha. Figure 6(c) showed the spatial variation of TN loading
the HTRW. The TN loading varied interval from 0.001 to 365.36 kg/ha. The TN loading had
the same spatial characteristics with TP loading. The loading output intensity of TN in the
Daliao River watershed was greater than that in the Hunhe River watershed, and the Taizi River
watershed was the lowest. Large amounts of fertilizer were applied in the study area. Nitrate
and organic N accounted a substantial portion of the fertilizer used in HTRW. Therefore, the
loading output intensity of TN in the watershed was very high. The regions with a great
proportion of farmland, such as the middle & lower reaches of the Hunhe River, the lower
reaches of the Taizi River and the tributaries in the upper reaches of the Daliao River, have high
output intensities of TN. The organic N contents in forested land was very low. Thus, the output
intensity of TN in regions with high vegetation cover of forest, such as the mountainous area in



the upper reaches of the Taizi and Hunhe rivers, was very low. The output loading intensity of
TN in the municipal districts with high developed area was the lowest, such as the municipal
districts of Fushun city and Shenyang city in the Hunhe River watershed, and the municipal
districts of Benxi city, Liaoyang city and Shenyang city in the Taizi River watershed.
The loading intensity of TN & TP in the HTRW were characterized by its regional
distribution. Although the counties of Qingyuan, Yibin and Benxi county, located in the upper
reaches of the HTRW, had high output of water and silt, their loading intensities of pollution
were not high. From the unit area perspective, the maximum loading intensities of TN & TP
were 365.36 & 259.83 kg/ha, respectively. The regions with high loading intensities of TN &
TP were mainly distributed in Taian, Haicheng, and Fushun city. The loading intensities of TP
& TN near the Dahuofang, Tanghe, Shenwo and Tanghe reservoirs were not high, ranging from
0.006 to 9.584 kg/ha, from 0.08 to 19.485 kg/ha, respectively. Based on the topography and soil
type distribution, the gradient in the upper stream of HTRW was usually high. The soil type is
predominately brown soil and salted paddy soil, both of which are easily eroded. The
topography in the lower reaches is usually even, as in the cities of Anshan, Haicheng, Yingkou
and Panjin. The elevation is not high, and the soil type is usually predominately meadow soil
and brown soil, both of which have a higher soil erosion rate, silt loss and loading intensity of
pollutants. The regions with heavy loading intensities of TN & TP included Xinmin county,
located in the middle and lower reaches of the HTRW, the municipal district of Shenyang city,
Liaozhong county, Dengta city, Liaoyang county, the municipal district of Anshan city,
Haicheng city and a portion of Dashiqiao city. Based on the land development pattern in the
Taizi River, dry fields and paddy fields were mainly distributed on the plain area of this
watershed, which is therefore a core source of loading intensity. The spatial difference in the





loading intensity between TN & TP were inconspicuous. Based on the topography, landform,
soil types and land development status in the watershed, the upper stream of watershed have
high vegetation coverage, less farmland and a low loading intensity of pollutants; the lower
reaches of the watershed have more farmland, high rates of fertilizer application and a high soil
erosion and pollution loading (Yin et al.,2011). To sum up, the spatial characteristics of TN
loading was the result of comprehensive effect from precipitation/run off characteristics, soil
properties, soil erosion and vegetation coverage. Therefore, in order to effectively control TN
loading and soil erosion in the HTRW, the BMPs, fallow measures of cultivated fields,
watershed vegetation restoration and soil & water conservation in the upper stream, which were
the most important measure that should be implemented.

## 412  3.3 NPS pollutant loading under EPS

The prevalence of farmland within a watershed has long been an important question, and
strong evidence exists of a correlation between land development mode and water environment
protect & rehabilitation at the basin scale. Numerous studies have used land use data and
stepwise regression analysis to explore relationships between land use and water quality
parameters and ecological integrity on a regional scale, including sub-basins, river riparian
buffer zones, and specific monitoring sites (Uriarte et al., 2011; Schiff, 2007; King et al., 2005).
The riparian buffer zones could effectively reduce the concentration levels of $NO_3^-$ in water,
which was 47% lower than the soil content (Venkatachalam et al.,2005). The dry farmland
caused a higher NPS pollutant loading, followed by paddy, rural and urban area, forest land,
and shrub land. Under this developmental scenario, the area of farmland in the watershed was
reduced; a modest area of farmland (29 500 ha, accounting for 2.74 % of the total farmland area)
was converted to forestland (included shrub land, 14 753 ha), grassland (5 899 ha), wetland (8





848 ha); and NPS pollution from farmland decreased. The objective of water quality protection
within the critical zoning of the watershed was realized. For the riparian buffers can be planted
in various diverse vegetation, the N removal rate of 60m wide woody soil buffer zone was 16%
and 38% higher than that of shrubbery and grassland, respectively (Aguiar et al.,2015). Urban
& rural areas were considered as the same type of land use in SWAT, about 1 kilometer within
both banks of the tributaries of the Hunhe, Taizi and Daliao rivers and 5 kilometers surrounding
reservoirs were defined as buffer zones, including 1946 km$^2$ of farmland, urban land, and rural
residential land, which accounts for 7.1 % of the total area in the watershed. The woodland coverage
rate was reduced by 1%, the loading intensity of sediment, TP & TN increased by 0.01~11.34, 0.15-
2.83, and 0.40-14.00 kg/km$^2$, respectively. The output of pollutant production under EPS was
calculated by transforming the existing land use type.
Based on the parameter quantification results of SWAT, the TN & TP losses from farmland
was effectively reduced after the modification of the land use structure. TN & TP respective
range of change was from 0 to 365.357 kg/ha, and from 0 to 259.834 kg/ha. The annual losses
of TN & TP were reduced by 13 839 and 1 946 t/a, respectively. In comparison, the output of
NPS pollutant production under the EPS was decreased by 21.9% compared with that under the
status quo scenario, whereas the outputs of TP & TN were reduced by 10.4% and 25.9%,
respectively. Under EPS, the average loading intensities of TN & TP were 14 and 6 kg/ha on a
unit area basis, which were 14.3% and 26.3% less than the loading intensities under status quo
scenario. The NPS pollution loading decline obviously in the EPS. The variation of TP & TN
pollution loading between status quo and EPS was shown in Table 2. The amount change
indicated that riparian buffer and land development pattern change could effectively reduce the
NPS pollutant loading in the HTRW.



## 4. Conclusions

The NPS pollution is prone to cause in dry farmland, paddy, rural & urban areas. The SWAT model has been applied to study NPS in China by numerous research literature, they were mainly focuses on scenario simulation of NPS pollution and management in agricultural areas with rich hydrological and meteorological data. The basic monitoring data of HTRW were deficient, we selected the SWAT as the feasible method to access NPS pollutant loading in watershed level. We applied certain practices based on EPS to reduce the NPS pollutant loading in the Hunhe River, Taizi River and Daliao River watershed. The status quo scenario and EPS were used to calculate the output of NPS pollutant production. Under the status quo scenario, the soil erosion modulus in the HTRW was 0.811 t/ha, and the soil erosion in the Daliao River watershed was the most severe. The TP & TN annual loading in the HTRW was 19, and 7 kg/ha, respectively. In the middle and lower stream of HTRW has a higher NPS pollutant loading, which included the urbanization and population density highly region of Shenyang, Anshan and Liaoyang. Under the EPS, the TN & TP per unit area were 14, and 6 kg/ha, respectively. The output of NPS pollutant production, the loading intensities of TN & TP was reduced by 21.9%, 25.9% and 10.4% compared with the status quo scenario, respectively. In different regions of NPS pollutant loading in the HTRW changes greatly, and the pollutant loading intensity of different nutrients in the same region is slightly different. Land eco-restoration and land development mode adjustment measures should be practiced to reduce NPS pollutant loading of cultivated land.

The SWAT model can be used to calculate and access the source, and potential reduction of agricultural NPS pollutants based on different land use type. The reliability of SWAT evaluation results is decided by information completeness and the reasonable degree of





parameter initialization. In HTRW some data were missing, such as the rainfall intensity, and
water pollution, et al. The data inaccuracy and local factors has a certain impact on SWAT
model accession result. To determine the pollutant reduction under different land development
patterns, and examine uncertainty of sensitivity parameters, SWAT model in China has wide
range of potential application.
**Acknowledgements** The study was financially supported by the National Key Research and
Development Program of China (2016YFC0401408) and Project Based Personnel Exchange
Program with China Scholarship Council & German Academic Exchange Service of 2015. The
author appreciates the experts & scholars of Helmholtz Centre for Environmental Research –
UFZ (Leipzig, Germany), as well as anonymous reviewers for their valuable comments and
criticisms.

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



**Figure captions:**


**Figure 1.** Basic information on the HTRW. The figure has been supplied by www.geodata.cn,
which is a national science and technology basic conditions platform and an earth system
science data sharing platform. The figure information is public. The Liaoning province Water
Resources Administrative Bureau granted permission for the basic information in the HTRW.
**Figure 2.** Data information in the HTRW.
**Figure 3.** Parameters calibration of SWAT model in the HTRW
**Figure 4.** The stream flow validation result of typical monitoring station.
**Figure 5.** The nutrients validation in Beikouqian station.
**Figure 6.** NPS pollution loading distributions of HTRW under status quo scenario.





**Table 1.** The pollutant production in the HTRW under status quo scenario

| Watershed | Area (km²) | Run off (E+08 m³) | Pollutant (t) | | | Pollutant loading (kg/ha) | | |
|---|---|---|---|---|---|---|---|---|
| | | | Sediment | TP | TN | Sediment | TP | TN |
| Hunhe River | 11 565 | 24.04 | 220 004 | 8 993 | 24 264 | 190 | 8 | 21 |
| Taizi River | 13 903 | 33.31 | 1 699 996 | 6 399 | 19 010 | 1 223 | 5 | 14 |
| Daliao River | 1 913 | 1.60 | 300 002 | 3 315 | 10 048 | 1 568 | 17 | 53 |
| Total/Average | 27 381 | 58.95 | 2 220 002 | 18 707 | 53 322 | 811 | 7 | 19 |

Source: China Hydrology; National earth system data sharing infrastructure; Field investigation of Liaoning province; Chemical fertilizer/Land area/Soil erosion statistics yearbook of Liaoning province; Liaoning province bureau of Meteorology.

**Table 2.** The variation of TP & TN pollution loading between EPS and status quo scenario

| Watershed | Pollutant loading of EPS (kg/ha) | | Pollutant loading variation (kg/ha) | | Farmland variation (ha) | Forestland variation (ha) | Grassland variation (ha) | Wetland variation (ha) | Pollutant annual variation(t/a) | |
|---|---|---|---|---|---|---|---|---|---|---|
| | TP | TN | TP | TN | | | | | TP | TN |
| Hunhe River | 7 | 16 | -1 | -5 | -12460 | +6231 | +2492 | +3737 | -838 | -5743 |
| Taizi River | 4 | 10 | -1 | -4 | -14979 | +7491 | +2995 | +4493 | -776 | -5606 |
| Daliao River | 16 | 40 | -1 | -13 | -2061 | +1031 | +412 | +618 | -332 | -2490 |
| Total/Average | 6 | 14 | -1 | -5 | -29500 | +14753 | +5899 | +8848 | -1946 | -13839 |

"—" denotes a decrease compared to status quo scenario; "+" denotes an increase compared to status quo scenario.



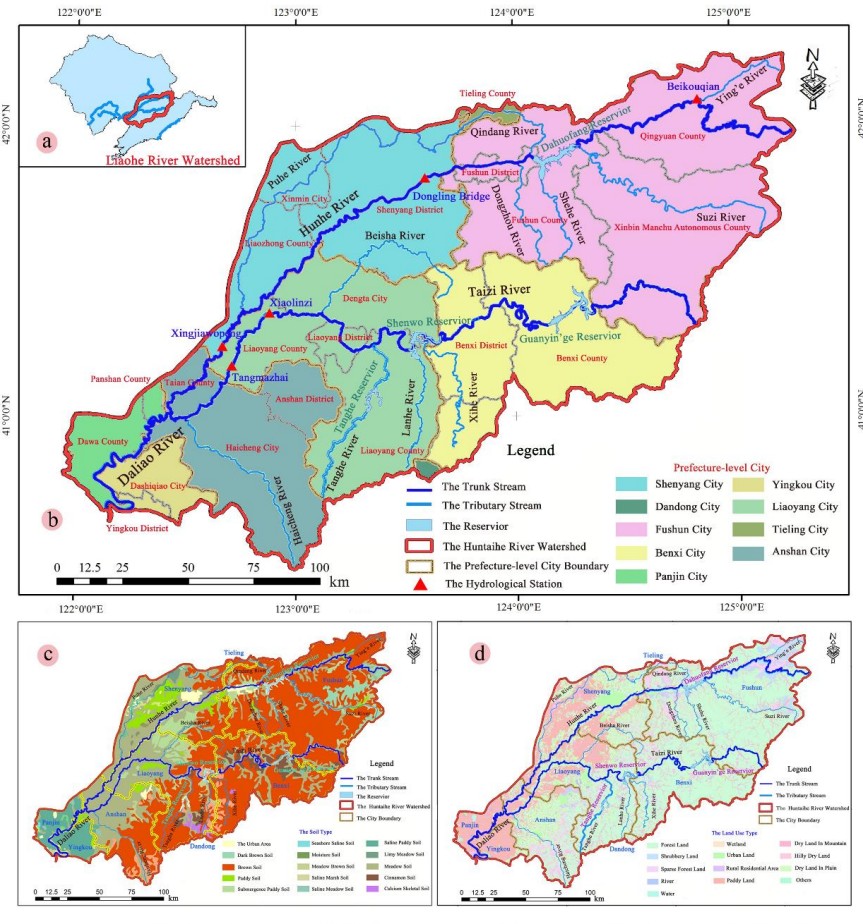

a. The location of the HTRW.

b. The geographical zoning of HTRW.

c. The land use type of HTRW.

d. The soil type of HTRW.

**Figure 1.** Basic information on the HTRW. The figure has been supplied by www.geodata.cn, which is a national science and technology basic conditions platform and an earth system science data sharing platform. The figure information is public. The Liaoning province Water Resources Administrative Bureau granted permission for the basic information in the HTRW.





| Data type | Data scale | Data description | Data source | |
|---|---|---|---|---|
| DEM (Digital Elevation Model) | 1:250 000 | Elevation, overland and channel slopes and lengths | Institute of Geographical and Natural Resources Research; Chinese Academy of Sciences; National Geomatics Center of China | Input |
| Land use | 1:100 000 | Land use classifications | Institute of Geographical and Natural Resources Research; Chinese Academy of Sciences | Input |
| Soil properties | 1:1000 000 | Soil physical and chemical properties | Institute of Soil Science; Chinese Academy of Sciences | |
| Meteorological data | / | Precipitation, daily maximum and minimum air temperature, relative humidity and solar radiation | China Meteorological Administration; Liaoning province bureau of Meteorology | Input |
| water quantity and quality | / | / | Local hydrographical station and environmental monitoring station | Input |
| Social economic data | / | Population, livestock rearing, fertilizer application | Field investigation; Statistics yearbook | Input |

SWAT of HTRW

**Figure 2.** Data information in the HTRW.





| Parameter | Descriptions of parameter | Value bounds | Default value | Calibrated value | Sensitive parameters |
|---|---|---|---|---|---|
| CN2 | Initial SCS Runoff curve number for moisture condition | 25-92 | — | Reduce by 5 | Soil water content |
| ESCO | Soil evaporation compensation factor | 0.01-1 | 0.95 | 0.19 | Surface runoff simulation |
| GWQMN | Threshold depth of water in shallow aquifer required for the return flow to occur | 0-5000 | 0 | 1200 | Surface runoff simulation |
| SPEXP | Exponent parameter for calculating sediment entrained in channel sediment routing | 1-1.5 | 1 | 1.45 | Sediment assessment |
| PRF | Peak rate adjustment factor for sediment routing in the main channel | 0-2 | 1 | 1.97 | Surface runoff simulation |
| SURLAG | Surface runoff lag time | 1-24 | 4 | 4 | / |
| ADJ_PKR | Peak rate adjustment factor for sediment routing in sub basins | 0.5-2 | 0.5 | 2 | Phosphorus assessment |
| PPERCO | P percolation coefficient | 10-17.5 | 10 | 17.5 | Phosphorus assessment |
| PHOSKD | P soil partitioning coefficient | 100-200 | 175 | 175 | / |
| P_UPDIS | P uptake distribution factor | 0-100 | 20 | 80 | / |
| PSP | Phosphorus sorption coefficient | 0.01-0.7 | 0.4 | 0.6 | / |
| SOL_LABP | Initial soluble P concentration in surface soil layer (mg/kg) | 0-100 | 0 | 12 | Phosphorus assessment |
| NPERCO | N percolation coefficient | 0-1 | 0.2 | 0.8 | Nitrogen assessment |
| SOL_NO₃ | Initial NO₃ concentration in the soil (mg/kg) | 0-100 | 0 | 20 | Nitrogen assessment |

**Figure 3.** Parameters calibration of SWAT model in the HTRW.

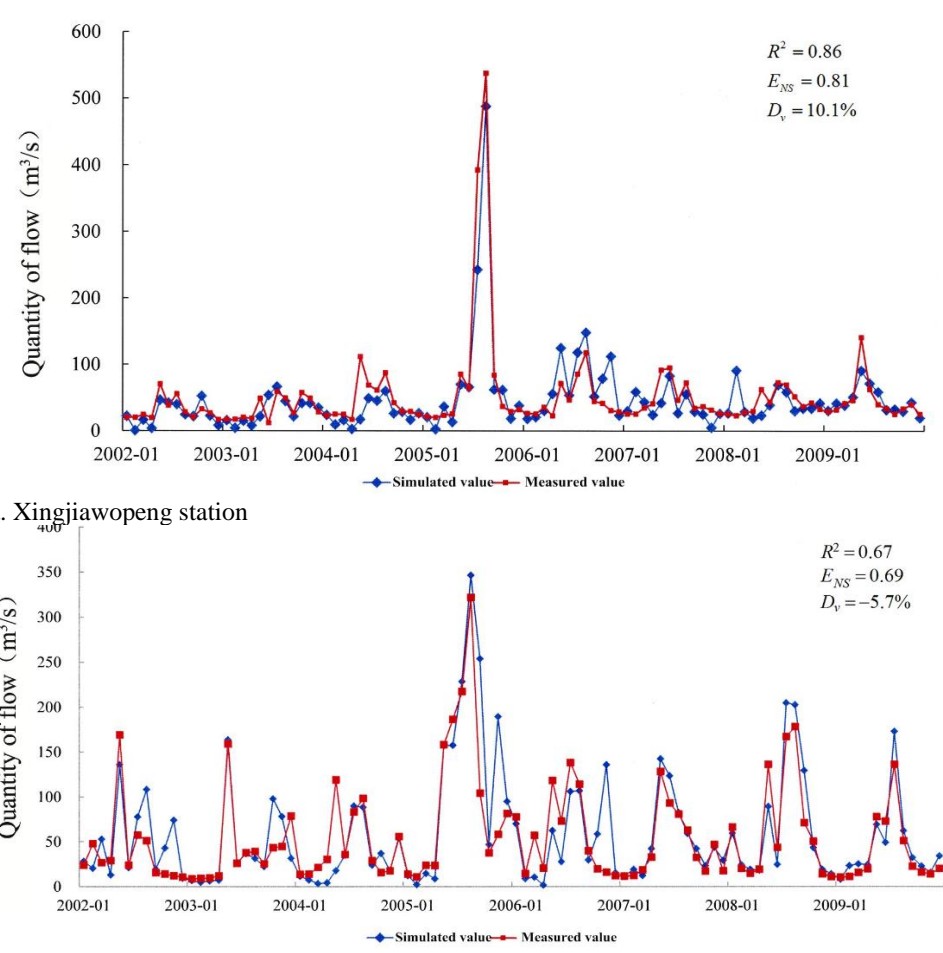

a. Xingjiawopeng station

b. Tangmazhai station

**Figure 4.** The stream flow validation result of typical monitoring station.




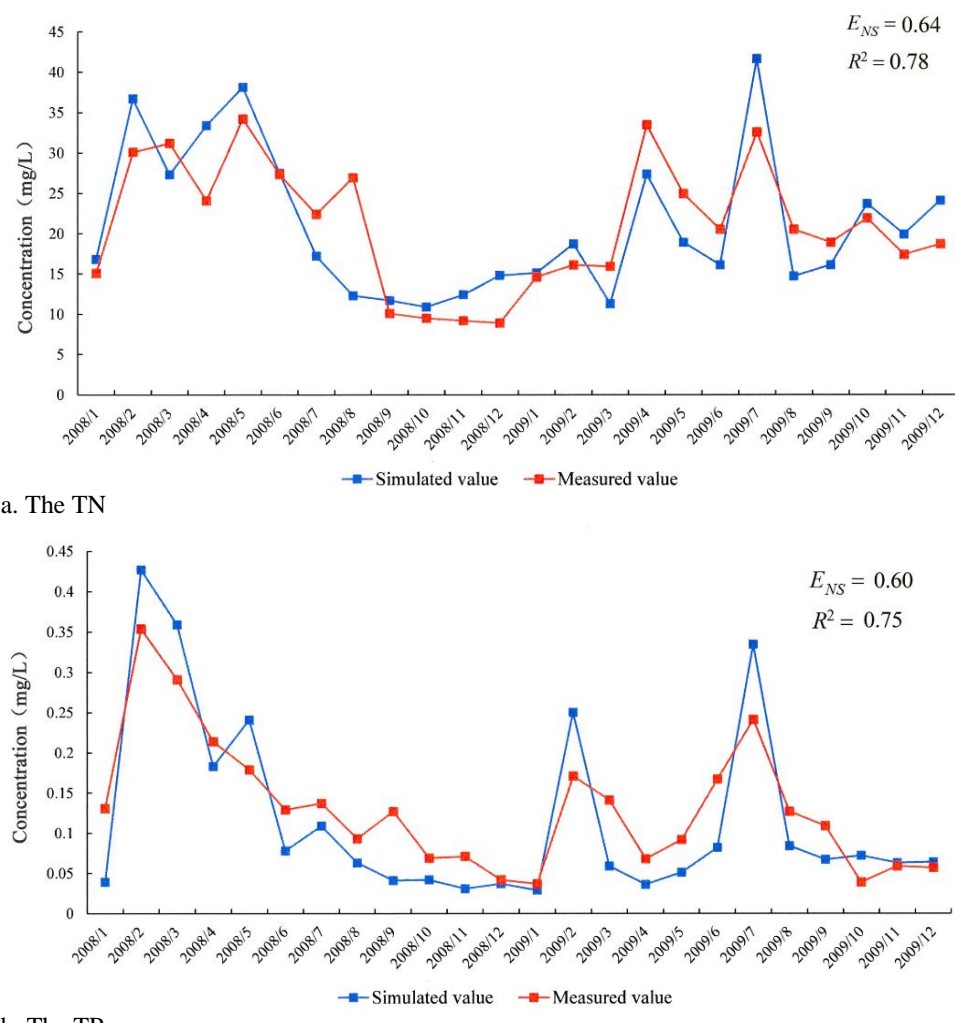

a. The TN

b. The TP

**Figure 5.** The nutrients validation in Beikouqian station.





a. The sediment.

b. The TP.



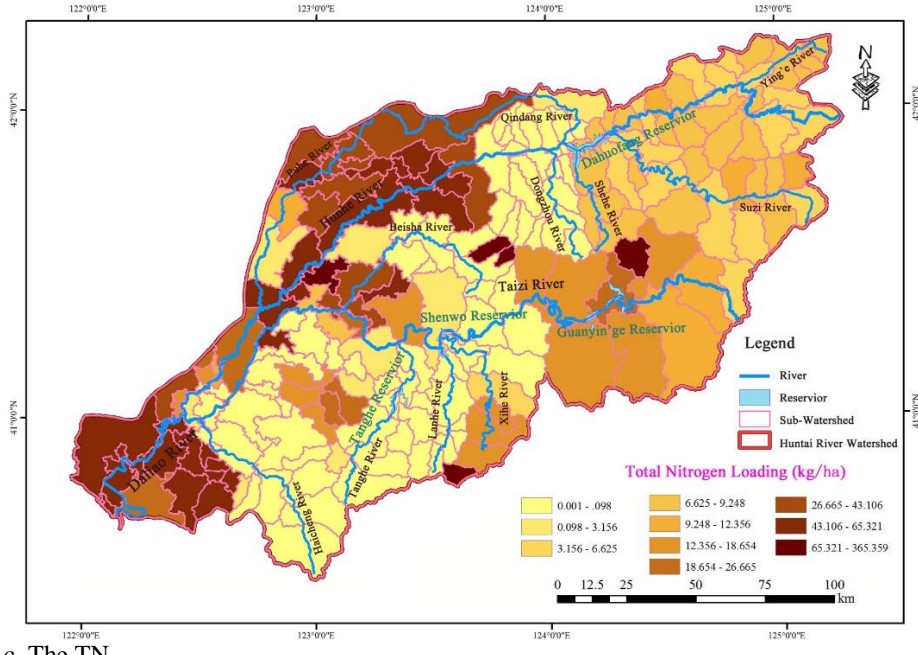

c. The TN.

**Figure 6.** NPS pollution loading distributions of HTRW under status quo scenario.