# Peer review of "Reduction Assessment of Agricultural Non-Point Source Pollutant"

_Hydrology and Earth System Sciences, 2017_

## Author Comment (AC1) · 11 Jan 2018

Dear editors, We would like to submit the enclosed manuscript entitled "Reduction Assessment of Agricultural Non-Point Source Pollutant Loading" (No.: hess-2017-755), which we wish to be considered for publication in "Hydrology and Earth System Sciences" (HESS). No conflict of interest exits in the submission of this manuscript, and manuscript is approved by all authors for publication. We would like to declare on behalf of my co-authors that the work described was original research that has not been published previously, and not under consideration for publication elsewhere, in whole or in part. All the authors listed have approved the manuscript that is enclosed. In this work, we evaluated the manuscript is a part of our present research achievement, and which is a good paper. I hope this paper is suitable for "Hydrology and Earth System

Sciences (HESS)". We submitted our manuscript "Reduction Assessment of Agricultural Non-Point Source Pollutant Loading", Reference hess-2017-716 to "Hydrology and Earth System Sciences" on 6th Dec., 2017. Non-point source (NPS) pollution has become the largest threat to water quality, in recent years. With the development of technology, application of models to control NPS pollution has become a very common practice for the management of soil and water resources on watershed scale in China. The Soil and Water Assessment Tool (SWAT) is a semi-distributed model, that was primarily developed to estimate the impacts of various land use and management practices on water, sediment, and agricultural chemical yields on water quantity and water quality in complex watersheds. Based on the overview of published papers with application of SWAT, the study topics is mainly focus on nutrients, sediments and related BMPs, impoundment and wetlands, hydrologic characteristics, climate change impact, and land-use change impacts. A SWAT model was constructed based on rainfall runoff and land use type. The migration-transformation processes of agricultural NPS pollutants were simulated and calculated based on the SWAT model. Besides, the loadings and distribution traits of NPS pollutants were also systematically analyzed based on the model. The model was used to quantify the spatial loading intensities of NPS Nutrient (Total nitrogen-TN and Total phosphorus-TP) to Huntai River Watershed (HTRW, Liaoning province, China) under two scenarios (without and with buffer zones). The SWAT model was calibrated and validated using actual monitoring data as well as the physical properties of the underlying substrate, hydrology, meteorology and pollutant sources in the HTRW. Scenario settings are mainly based on the changes of surface runoff and sediments, climate and land-use change from different spatial scales, and climatic/physiographic zones. About 1 km within both banks of the trunk streams of the Huntai, Taizi and Daliao rivers, and 5 km surrounding the reservoirs were defined as buffer zones. Existing land use type within the buffer zone was changed to reflect the natural environment. The output of pollutant production under the "environmental protection" scenarios (EPS) was calculated based on the status quo scenario. Under the status quo scenario, the annual mean modulus of soil erosion in the HTRW was 811

kg/ha, and the output intensities of TN and TP were 19 and 7 kg/ha, respectively. For the unit area, the maximal loading intensities for TN and TP were 365.36 and 259.83 kg/ha, respectively. In terms of spatial distribution, TN and TP loading varied substantially. Under the EPS, the magnitude of the nitrogen and phosphorus losses from cultivated land decreased to a certain degree, and the TN and TP pollution loading per unit area were reduced by 5 and 1 kg/ha annually, respectively. In comparison, the quantity of NPS pollutant production under the EPS was reduced by 21.9% compared with the status quo scenario, and the quantities of TP and TN decreased by 10.4% and 25.9%, respectively. These changes suggested a clear reduction in the export loading of agricultural NPS pollution. Loading intensities analysis showed that land use type is one key factor for controlling NPS pollution. The NPS pollution loading decreased under the EPS, which showed that environmental protection measure could effectively cut down NPS pollution loading in HTRW. SWAT was used to assess the reduction of agricultural NPS pollutant. However, SWAT model requires a large amount of data about the watershed being modeled; the data inaccuracy and local factors would impact the accuracy of the SWAT model. Further research is required to recognize the main factors that affect the accuracy of different NPS pollutants loading, examine uncertainty of sensitivity parameters, and extend the potential application range of SWAT in China. And the Highlights of the paper were, • SWAT was used to assess the reduction of agricultural NPS pollutant. • Buffer zone of land use type could reflect the natural environment. • 21.9% pollutant reduction under the EPS. SWAT model was simulated and calculated migration-transformation processes of agricultural NPS pollutants. Existing land use type within the buffer zone was changed to reflect the natural environment. The quantity of NPS pollutant production under the EPS was reduced by 21.9% compared with the status quo scenario. SWAT model requires a large amount of data about the watershed being modeled. We hope you are interested in the research content of this article and can handle this article. As the corresponding author of the paper, I would greatly appreciate if you could inform me of anything about it. On behalf of my co-authors, we would like to express our great appreciation to you.

We wish you a happy work. Thank you very much for your time and consideration. If you have any further questions, please do not hesitate to contact me.

Yours sincerely, Yicheng Fu(corresponding author), Wenbin Zang, Jian Zhang, Hongtao Wang, Chunling Zhang, & Wanli Shi E-mail address of corresponding author: swfyc@126.com (Y.C. Fu).

11th, Jan. 2018

Please also note the supplement to this comment:
https://www.hydrol-earth-syst-sci-discuss.net/hess-2017-755/hess-2017-755-AC1-supplement.pdf

---

## Short Comment (SC1) · 12 Jan 2018

Generally, the manuscript addresses an important topic. The work in the manuscript is sufficient to be a publication. However, the writing needs to be improved in some sections of the manuscript. Please see specific comments below. Abstract: Please write full words of abbreviations before using them. For example NPS, SWAT in the abstract. The authors should check abbreviations throughout the manuscript. L16: "The study topics is mainly focus on", correct to "The study topic mainly focuses on". The purpose of the study is very general. I prefer specific objectives of the study. L17-18: " SWAT model was constructed based on rainfall runoff 18 and land use type": SWAT model also uses soil types and slope information. L20: What do you mean by systematically analyzed? Can you describe what you did? L24: What you mean by

"scenario settings" in your study? In the Results and Discussion of the abstract, you should mention your results for calibration and validation before discussing about the results from scenarios.

Introduction L53-54: "The concentrate...between different areas". Grammar is not right. Please rewrite. L73-L74: "The SWAT model has 701 mathematical equations..." This is really unnecessary. The model is continuously updated and equations are continuously added. L91: "contrast the different" I guess you mean compare.

Materials and Methods Section 2.1 about description of study area is too long. Please shorten it and only mention necessary information. L141-L147 " For the calculation process ... farmers status quo". I think these sentences should belong to the model setup section. The description about SWAT model is too long. Since we can find these information in many previous studies and in the manual of SWAT, there is no need to describe them in details. Please shorten it and only choose the necessary information to describe. L184-185: " We used $30\times30$ grid data (elevation) as the basis for DEM operation". What did you do to prepare the DEM data? L193-195 " The database of the underlying substrate was constructed based on the database of soil types using the soil properties & land development data as underlying substrate parameters". I don't understand what you want to say here. What are substrate parameters here? L204-205 "All the data were validated by the standard procedures used by the SWAT". Can you specify the standard procedures? L228-229: Which period is used for calibration, and for validation? L283-288: Your description on streamflow calibration is not clear about how you did for annual calibration and how you used the annual calibration to do monthly calibration. Did you use SWAT-CUP for this calibration?

Is the SWAT setup you used for calibration called the status quo scenario described in the Scenarios setting? L271-272: 29 smaller modeling units, are they subbasins in SWAT? Or HRUs? Then after that you mentioned 184 HRUs. But with the number of soil types (26 types) and land use types (27 types), the number of HRUs (184) seems to be a very small number.

I think the results are valuable, however, I don't feel they have been presented well to the reader.

Conclusion I feel that the conclusion is just repetition of the results and discussion. I don't think you should repeat the number of TN and TP loads under two scenarios. You should summarize what you learn from the results and discuss about them.

Please also note the supplement to this comment:
https://www.hydrol-earth-syst-sci-discuss.net/hess-2017-755/hess-2017-755-SC1-supplement.pdf

---

## Author Comment (AC2) · 15 Jan 2018

Dear editors, We would like to submit the enclosed manuscript entitled "Reduction Assessment of Agricultural Non-Point Source Pollutant Loading" (hess-2017-755), which we wish to be considered for publication in "Hydrology and Earth System Sciences (HESS)". No conflict of interest exits in the submission of this manuscript, and manuscript is approved by all authors for publication. In this work, we evaluated the manuscript is a part of our present research achievement, and which is a good paper. I hope this paper is suitable for "HESS". The main point our thesis wishes to address is to reflect on the practical application of and the solutions provided by the SWAT models in relation to China's sparse hydrological basin information; to provide point by point model constructions; an explanation of our process; an analysis of our

results, and the expansion of the utilization of the SWAT model from an advanced and disciplined perspective. SWAT was used to assess the reduction of agricultural NPS pollutant. Buffer zone of land use type could reflect the natural environment. 21.9% pollutant reduction under the EPS. We have tried our best to revise the manuscript to hope to meet with approval. The manuscript has been thoroughly checked again and revised as suggested with the help of an English teacher. It is believed that the revised paper will be readable and could meet the standard generally for publication. Thank you very much for your consideration and help. Looking forward to hearing from you soon. Thank you very much for your time and consideration.

Yours sincerely, Dr. Yicheng Fu E-mail: swfyc@126.com +86-10-68781880 (office); +86-10-68572778 (fax) Organization name: China Institute of Water Resources and Hydropower Research (IWHR) Organization address: A-1 Fuxing Road, Haidian District, 100038 Beijing

15th, Jan. 2018

Please also note the supplement to this comment:
https://www.hydrol-earth-syst-sci-discuss.net/hess-2017-755/hess-2017-755-AC2-supplement.pdf

―――――――――――――――――――

---

## Author Comment (AC3) · 15 Jan 2018

Dear Editor, We are so appreciated for your letter on our manuscript "Reduction Assessment of Agricultural Non-Point Source Pollutant Loading", Reference No: hess-2017-755. We are also extremely grateful to the editors'/reviewers' comments on our manuscript and carefully considered every comment, and made cautious revision accordingly. Based on their suggestions, we have answered the questions in detail one by one. If you have any other questions about this paper, I would quite appreciate it if you could let me know them in the earliest possible time.

Most sincerely,

YiCheng Fu, Wenbin Zang, Jian Zhang, Hongtao Wang, Chunling Zhang, and

Wanli Shi First Contact: Yicheng Fu, swfyc@126.com Second Contact: Jian Zhang, zhangjian5love@163.com

Corresponding author: Name: yi-cheng FU E-mail: swfyc@126.com 15th, Jan. 2018

Please also note the supplement to this comment:
https://www.hydrol-earth-syst-sci-discuss.net/hess-2017-755/hess-2017-755-AC3-supplement.pdf

**Supplement:**

**Dear Editor,**

We are so appreciated for your letter on our manuscript "Reduction Assessment of Agricultural Non-Point Source Pollutant Loading", Reference No: hess-2017-755. We are also extremely grateful to the editors'/reviewers' comments on our manuscript and carefully considered every comment, and made cautious revision accordingly. Based on their suggestions, we have answered the questions in detail one by one. If you have any other questions about this paper, I would quite appreciate it if you could let me know them in the earliest possible time.

Most sincerely,

YiCheng Fu, Wenbin Zang, Jian Zhang, Hongtao Wang, Chunling Zhang, and Wanli Shi
First Contact: Yicheng Fu, swfyc@126.com
Second Contact: Jian Zhang, zhangjian5love@163.com

Corresponding author:

Name: yi-cheng FU

E-mail: swfyc@126.com

15th, Jan. 2018

**Additive list**

We have studied the valuable comments from you, the assistant editor and reviewers carefully, and tried our best to revise the manuscript. The point to point responds to the reviewer's comments are listed as following.

**Reviewer's Responses to Questions**

Generally, the manuscript addresses an important topic. The work in the manuscript is sufficient to be a publication. However, the writing needs to be improved in some sections of the manuscript. Please see specific comments below.

(1) Please write full words of abbreviations before using them. For example, NPS, SWAT in the abstract. The authors should check abbreviations throughout the manuscript.

**Answer:** Thanks for your very thoughtful suggestion.

We have made serious changes to the expression of abbreviations in the whole paper, such as NPS (Non-point source), SWAT (Soil and Water Assessment Tool), TN (Total Nitrogen), TP (Total Phosphorus), HTRW (Huntai River Watershed), environmental protection scenario (EPS), DEM (Digital Elevation Model), and BMPs (Best Management Practices scenarios).

The revised contents could be found in the file of "paper revised version (clean)".

(2) L16: "The study topics is mainly focus on", correct to "The study topic mainly focuses on". The purpose of the study is very general. I prefer specific objectives of the study.

**Answer:** Thanks for your very thoughtful suggestion.

We have revised the "The study topics is mainly focus on" to "The study topic mainly focuses on".

In order to make the article clear, we have revised the ""The study topic" to "The focus point". This section is the application scope of SWAT model, which was not the specific objectives of the study. The study objectives of the paper was "The model was used to quantify the spatial loading intensities of NPS nutrient TN (Total Nitrogen) and

TP (Total Phosphorus) to HTRW (Huntai River Watershed) under two scenarios (without & with buffer zones). The NPS pollutant loading decreased under the EPS, which showed that environmental protection measure could effectively cut down NPS pollutant loading in HTRW. SWAT was used to assess the reduction of agricultural NPS pollutant."

The revised contents could be found in the file of "paper revised version (clean)" & paper revised version (with track changes).

(3) L17-18: " SWAT model was constructed based on rainfall runoff and land use type": SWAT model also uses soil types and slope information.

**Answer:** Thanks for your very thoughtful suggestion.

We have improved SWAT model information, and have added the soil types and slope information to the SWAT. The revised contents could be found as the followed,

"SWAT model was constructed based on rainfall runoff, land use type, soil types and slope information.".

(4) L20: What do you mean by systematically analyzed? Can you describe what you did?

**Answer:** Thanks for your very thoughtful suggestion.

The systematically analysis contained three parts, which were (1) scenarios setting of SWAT; (2) modelling validation of SWAT in HTRW; (3) NPS pollutant loading calculation under status quo scenario & EPS.

The revised section was as followed,

Besides, the loadings and distribution traits of NPS pollutants were also systematically analyzed based on the model (scenarios setting, modelling validation, and pollutant loading calculation under status quo scenario & EPS).

(5) L24: What you mean by "scenario settings" in your study?

**Answer:** Thanks for your very thoughtful suggestion.

The "scenario settings" is the mean of "Land use types differences".

The revised contents could be found in the file of "paper revised version (clean)" & paper revised version (with track changes).

(6) In the Results and Discussion of the abstract, you should mention your results for calibration and validation before discussing about the results from scenarios.

**Answer:** Thanks for your very thoughtful suggestion. We added the following contents,

The $E_{NS}$ (Nash-Sutcliffe efficiency coefficient) & $R^2$ (certainty coefficient) of stream & nutrients (TN & TP) in typical hydrological station were both greater than 0.6, and the $|Dv|$ (relative deviation) was less than 20%. The SWAT could be used in HTRW.

The revised contents could be found in the file of "paper revised version (clean)" & paper revised version (with track changes).

(7) Introduction, L53-54: "The concentrate…between different areas". Grammar is not right. Please rewrite.

**Answer:** Thanks for your very thoughtful suggestion.

We carefully devised the expression of the sentence. The revised contents were followed,

The NPS pollutant concentrate in water is dependent on the discharge intensity and pollutant treatment rate, therefore, which was difficult to make a fair comparison between different areas (Tucci 1998; Dingman 2002; de Oliveira et al.,2016).

(8) Materials and Methods. Section 2.1 about description of study area is too long. Please shorten it and only mention necessary information.

**Answer:** Thanks for your very thoughtful suggestion. We have shortened the length of Section 2.1. We only provided the necessary information of study area. The contents were been found as following,

The HTRW (40°27′~42°19′N, 121°57′~125°20′E) is in Liaoning province (Northeast China), and the watershed area is 2.73×104 km2, which takes about 1/5 of the area of Liaoning province (Fig 1). The HTRW is a tributary of Liaohe River Basin (The Liaohe River Basin is one of China's larger water systems) and is consist of Hunhe River, Taizi River, and Daliao River. The Hunhe River, Taizi River, and Daliao River watershed is HTRW's sub-watershed. The HTRW has varied topography, low mountain is located in eastern part, and the other parts are alluvial plain. The elevation of northeast region is high. Loamy soils are mainly distributed in alluvial plain, and the average grade of lower HTRW is about 7%. HTRW area includes the cities of Fushun, Shenyang, Benxi, Liaoyang, Anshan, and Yingkou, most of Panjin city, some portions of Tieling city and a minor portion of Dandong city. The stream flow and nutrient were validated based on the five monitoring stations, Beikouqian, Dongling Bridge and Xingjiawopeng are located in Hunhe River, Xialinzi and Tangmazhai are in Taizi Rive. HTRW has temperate continental climate, the average annual temperature is 7°C, and precipitation is 748 mm.

(9) L141-L147 " For the calculation process … farmers status quo". I think these sentences should belong to the model setup section.

**Answer:** Thanks for your very thoughtful suggestion. We have put the " For the calculation process … farmers status quo" to the model setup section.

(10) The description about SWAT model is too long. Since we can find these information in many previous studies and in the manual of SWAT, there is no need to describe them in detail. Please shorten it and only choose the necessary information to describe.

**Answer:** Thanks for your very thoughtful suggestion. We have shortened the length of SWAT model description. We only provided the necessary information of SWAT model. We supplied some information of SWAT in the form of figure, such as Figure 1, and Figure 2.

(11) L184-185: " We used 30×30 grid data (elevation) as the basis for DEM operation". What did you do to prepare the DEM data?

**Answer:** Thanks for your very thoughtful suggestion.

We download the DEM data of HTRW from the SRTM (Shuttle Radar Topography Mission) data pack, the free data can be obtained on the website of http://srtm.csi.cgiar.org/SELECTION/inputCoord.asp. With GIS (Geographic Information System) platform, we obtained the DEM data of HTRW, as well as hydrological station & weather station distribution, by using the technology of DEM data projection transformation, splicing and cutting.

(12) L193-195 " The database of the underlying substrate was constructed based on the database of soil types using the soil properties & land development data as underlying substrate parameters". I don't understand what you want to say here. What are substrate parameters here?

**Answer:** Thanks for your very thoughtful suggestion.

The underlying substrate parameters means the data of topography characteristics, surface vegetation and soil types & distribution characteristics. These data were the basic to calculate NPS pollutant loading and distribution intensity changes.

(13) L204-205 "All the data were validated by the standard procedures used by the SWAT". Can you specify the standard procedures?

**Answer:** Thanks for your very thoughtful suggestion.

We added the related contents were as followed,

The SWAT uses the LH-OAT (Latin Hypercube One-factor-At-a-Time) sensitivity analysis method & SCE-UA (Shuffled Complex Evolution Algorithm) automatic calibration analysis method to determine the value of sensitive parameters.

The revised contents could be found in the file of "paper revised version (clean)" & paper revised version (with track changes).

(14) L228-229: Which period is used for calibration, and for validation?

**Answer:** Thanks for your very thoughtful suggestion.

We added the related contents were as followed,

The runoff, TN & TP loadings data used for calibration & validation were from 1992 to 2009, from 2006 to 2008, respectively.

In L287, to the stream flow, "For the simulation, 1990-1994 was the model preparation period, 1995-2001 was the model calibration period, and 2002-2009 was the model validation period." The contents could be found in the file of "paper revised version (clean)" (L296-L297).

In L304-306, to the nutrients, "Beikouqian, Xingjiawopeng, Xiaolinzi and Tangmazhai four hydrological stations had a continuous monthly water quality monitoring data from 2006 to 2007. Only the monthly data of TN & TP in Beikouqian were validated from 2008 to 2009 for the insufficient of water quality monitoring data.".
Therefore, the 2006-2007 was the model calibration period, and 2008-2009 was the model validation period.

The revised contents could be found in the file of "paper revised version (clean)" & paper revised version (with track changes).

(15) L283-288: Your description on streamflow calibration is not clear about how you did for annual calibration and how you used the annual calibration to do monthly calibration. Did you use SWAT-CUP for this calibration?

**Answer:** Thanks for your very thoughtful suggestion.

We added the related contents were as followed, (1) First, we dealt with the meteorological data and retained the 1990-2001 data series, then supplied the meteorological data simulation value from 1990 to 2001 by SWAT; (2) We input into the runoff data of 1995-2001 to SWAT-CUP model to calibrate the runoff parameters; (3) We took the (2) parameters into the database of SWAT, then extended the series of meteorological data to 1990-2009 and simulated runoff again.

(4) At last, we compared the runoff simulation values with monitoring value from 2002 to 2009.

The added contents could be found in the file of "paper revised version (clean)" & paper revised version (with track changes).

(16) Is the SWAT setup you used for calibration called the status quo scenario described in the Scenarios setting?

**Answer:** Thanks for your very thoughtful suggestion.

The scenarios setting for calibration was called the status quo scenario.

(17) L271-272: 29 smaller modeling units, are they sub-basins in SWAT? Or HRUs? Then after that you mentioned 184 HRUs. But with the number of soil types (26 types) and land use types (27 types), the number of HRUs (184) seems to be a very small number.

**Answer:** Thanks for your very thoughtful suggestion.

We added the related contents as followed,

To simulate the hydrological characteristics by SWAT, firstly, we divided the HTRW into a certain number of sub-basins according to DEM data, the sub-basins have the same characteristics of soil & land use; then we divided sub-basins into HRUs.

(18) I think the results are valuable, however, I don't feel they have been presented well to the reader.

**Answer:** Thanks for your very thoughtful suggestion.

In order to increase the readability of the paper, we reduced the number of pictures, and increased the number of tables to describe the reduction of agricultural NPS pollution loading. The spatial distribution of the mean annual TP and TN loading in the HTRW were 19, and 7 kg/ha, respectively. The region with a high NPS pollution loading is located in the middle and lower the HTRW, which included the urbanization and population density highly areas of Shenyang, Liaoyang and Anshan. Under the EPS, the TN and TP per unit area were 14, and 6 kg/ha, respectively. The output of NPS pollutant production, the loading intensities of TN & TP was reduced by 21.9%, 25.9% and 10.4% compared with the status quo scenario, respectively. The NPS pollution occurring within different sub-basins and regions located in the watersheds varied greatly, and the loading intensities of different pollutant types in the given sub-basin were slightly different. Land eco-restoration measures should be implemented to control agricultural NPS pollution from croplands. Therefore, SWAT simulation results provide a reference for the prevention of agricultural NPS pollution in agricultural watersheds.

(19) Conclusion

I feel that the conclusion is just repetition of the results and discussion. I don't think you should repeat the number of TN and TP loads under two scenarios. You should summarize what you learn from the results and discuss about them.

**Answer:** Thanks for your very thoughtful suggestion.

We have deleted the number of TN and TP loads under two scenarios. And summarized the contents that we learn from the results and discuss. We revised the contents as followed,

The NPS pollution is prone to cause in dry farmland, paddy, rural & urban areas. The SWAT model has been applied to study NPS in China by numerous research literature, they were mainly focuses on scenario simulation of NPS pollution and management in agricultural areas with rich hydrological and meteorological data. The basic monitoring data of HTRW were deficient, we selected the SWAT as the feasible method to access NPS pollutant loading in watershed level. We applied certain practices based on EPS to reduce the NPS pollutant loading in the Hunhe River, Taizi River and Daliao River watershed. The status quo scenario and EPS were used to calculate the output of NPS pollutant production. The output of NPS pollutant production, the loading intensities of TN & TP was reduced by 21.9%, 25.9% and 10.4% compared with the status quo scenario, respectively. In different regions of NPS pollutant loading in the HTRW changes greatly, and the pollutant loading intensity of different nutrients in the same region is slightly different. Land eco-restoration and land development mode adjustment measures should be practiced to reduce NPS pollutant loading of cultivated land.

The revised contents could be found in the file of "paper revised version (clean)" & paper revised version (with track changes).

We tried our best to improve the manuscript and made some changes in the manuscript. These changes will not influence the content and framework of the paper. And here we did not list the changes but marked in red in revised paper (Revision, changes marked).

We appreciate for Editors/Reviewers' warm work earnestly, and hope that the correction will meet with approval.

Once again, thank you very much for your comments and suggestions.

**Reduction Assessment of Agricultural Non-Point Source Pollutant Loading**

YiCheng Fu [a]*, Wenbin Zang[a], Jian Zhang[a], Hongtao Wang[b], Chunling Zhang[a], Wanli Shi[a]

[revised manuscript text omitted]

mmThe monsoon climate features of this watershed include an annual temperature ranging from 5-9°C and precipitation of approximately 748 mm.

The HTRW is in a conventional agricultural farming area, with a large area of farmland dominated by crop plants. The total area of farmland is 10 763 km$^2$ (account for 39.4% of the total area), including

4 086 km$^2$ of paddy field (dominated by rice) and 6 677 km$^2$ of dry farmland (including corn, soybean, vegetables and other crop plants). The upper reaches of the Hunhe and Taizi rivers have mountainous (69%), low hilly (6.1%) and plain land (24.9%). The economic output value of HTRW is dominated by agriculture.Agriculture is the main economic activity in HTRW. The farmland is mainly distributed in the floodplain area and valleys in riverine belts. Considering land patternBased on the land use, rainfall and source of pollutants, the HTRW faces a high risk of pollution from agriculture. Heavy use of fertilizers and soil erosion in the upper of HTRW has led to serious NPS pollution in HTRWThe massive application of fertilizers has caused the release of much N and P, resulting in serious NPS pollution in HTRW. For example, the Dahuofang reservoir of the Hunhe River and the water resources conservation area in its upper reaches are facing multiple threats, the agricultural NPS pollution is becoming increasingly serious and has not yet been controlled effectively (Shen et al., 2013c).

Fertilization in the HTRW is predominantly with nitrogen, followed by phosphorous and potassium. The heavy use of chemical fertilizers was mainly urea The types of fertilizer used in the watershed were mainly urea, diammonium phosphate and a small amount of potassium phosphate compound fertilizer. Atrazine and acetochlor were mainly used on dry farmland, and butachlor was mainly used in paddy fields. Based on the statistical data for 2006-2012, the quantity of fertilizers and pesticides applied in the watershed fluctuated annually. The upper reaches of the Huntai and Taizi rivers are dominated by mountains, the cultivation and harvesting of crops are conducted by hand, and therefore thorough statistics are not available.

At present, weeds and pests in farmlands were mainly controlled by pesticides and herbicides. The upstream is rich in forest resources, the downstream has a large number of farmland, special landscape layout makes the HTRW become potential area for agricultural NPS pollution.

.

[Figure]

a. The location of the HTRW.          b. The geographical zoning of HTRW.
c. The land use type of HTRW.         d. The soil type of HTRW.

**Fig_ure_ 1.** Basic information on the HTRW. The figure has been supplied by www.geodata.cn, which is a national science and technology basic conditions platform and an earth system science data sharing platform.

The figure information is public. The Liaoning province Water Resources Administrative Bureau granted permission for the basic information in the HTRW.

**2.2. Model description**

*2.2.1. The Soil and Water Assessment Tool*

SWAT is a semi-physical model developed to quantitatively calculate the response status of water quantity & quality to land use and management methods in the scale of watershed

(Gassman et al.,2007). SWAT is an effective to determine the long-term impact using monitoring data (Arnold et al.,2012). The basic data input for model running includes DEM/topography, soil type, vegetation status/ Land landscape, and best management practices scenarios

. The calculation unit of watershed SWAT model is sub-watershed, and HRU (Hydrological Response Units), the unit delineation is based on the underlying surface status, vegetation coverage, soil classification, and land use (Neitsch, 2005).

The HRUs of SWAT are automatically divided according to soil conditions, DEM, geomorphological features, and land development (Douglas-Mankin et al., 2010)

. For the calculation process is realized on HRU, therefore, we selected 0% land development, elevation/slope, and soil classification / attributes as the initial value on the scale of small key area

, therefore, 184 HRUs were delineated to determine NPS pollutant loading.

In order to assess pollutant loss and ecological flow status, the flow curve, soil nutrient loss curve, and water-salt balance equation were applied during the period of model debugging

. Meteorology data (sun radiation, atmospheric pressure, atmospheric temperature, precipitation and wind speed) were obtained from meteorological and hydrological stations of 12 cities located within HTRW.

precipitation, minimum and maximum temperature, solar radiation and wind speed) were obtained from 12 city weather stations located approximately within the watershed. The data of BMPs, such as crop sowing/harvest time, crop irrigation time, cultivation structure of cultivated land, fertilizer-use efficiency, and farmland planting plan were got from agriculture & environmental management department, or collected from the survey of farmers status quo. The farmland management information, such as the timing of manure and fertilizer application, crops harvest dates and land plantation structure were collected from detailed interviews with local farmers. Based on the above assessment results, we used QUAL2E (water quality model) to determine N & P yields loading, the route of sediment transport, and pollutant concentration of watershed outlet. The sediment, N and P yields from each sub watershed were subsequently routed through the channels to the watershed outlet, using the QUAL2E (water quality model) program.

   The SWAT is mainly used to assess the nutrient (N & P) production, migration, and transform.he SWAT model mainly simulates N and P cycling. These cycling processes occur simultaneously with the processes of the hydrological cycle and soil erosion. The N & P cycles simulation of SWAT was developed based on 5 different forms of N & 6 different forms of P, respectivelySWAT model's nitrogen and phosphorus cycles through five different pools of nitrogen (two inorganic forms: $NH_4^+$ and $NO_3^=$; three organic forms: fresh, stable and active) and six different pools of phosphorus (three inorganic forms: solution, active and stable; three organic forms: fresh, stable and active) in soil. The N & P cycles were consisted of the process of decomposition, mineralization, fixation, and conversionMineralization, decomposition, and immobilization are important processes in both N and P cycles. The NPS pollutant loading function is the basis of assessing N & P transport and transformationOrganic N and P transport with sediment is estimated using a loading function (McElroy, 1976; Williams et al.,1978; Zhang, 2005). Organic N & P losses calculation of SWAT was achieved by the integrated function of soil nutrient curve, NPS pollutant loading, soil properties change rate, and crop growth characteristicsDaily organic N and P runoff losses are calculated by loading functions based on the concentrations of these elements in the top soil layer, the sediment yield, and an enrichment ratio. The total amount of nitrate in lost soil was calculated by the product of water volume and nitrate concentration in water. Water volume is the consisted of surface runoff, groundwater runoff, and interflow/ subsurface flow. The concentration of soluble P in water is calculated by topsoil P stocks, runoff variation, ratio of soluble P, and soil particle characteristics.

Surface runoff from daily precipitation in HRU/Sub-watershed was calculated & assessed using the SCS-CN corresponding relationship curve and rainfall-runoff Coefficient (USDA Soil Conservation Service, National Engineering Handbook, 1972). With SCS-CN curve, saturated moisture, soil water profile/vertical distribution of soil moisture content, runoff module number of the underground water is determined, as well as the related parameters daily of precipitation.. The total discharge of runoff from sub-watershed/ HRUs is the sum of surface runoff flow, groundwater runoff flow, and interflow/ subsurface flow. Domestic water & irrigation water is direct consumptive water resources, the mainly water resources is surface runoff & groundwater runoff (Neitsch,2005). The main routing of water circulation in SWAT is network-node diagram and natural-artificial dualistic water cycle mode.  In the paper, we used a dualistic method for multi-layer and multi-function separation and interception of the rainfall and run off resources. Circulating flow of SWAT was varied with the dynamic changes of evaporation, infiltration, transport, and return flow (Arnold et al.,1998).

The HRUs of SWAT used soil erosion modulus, soil &

water loss coefficient, and Universal Soil Loss Equation (MUSLE) to analyze erosion and sediment yield (Williams, 1975)

We used 2009 version of SWAT to calculate the correlation parameters.

**2.2.2. Model inputs**

The data of DEM, geomorphology, underlying surface status, soil properties, land cover, meteorological

& hydrological data (precipitation, evaporation, temperature, and atmospheric pressure, et al.) were input to achieve the operation of SWAT (Niraula et al.,2013).

Table 1 supplied the basic data information to be used in SWAT model. We used $30 \times 30$

grid data (elevation) as the basis for DEM operation

. The DEM was selected as the topographical basis on which to construct the SWAT

model, to extract the scope of the study area and to construct the topographical model. The stream network in the study area was extracted using 1:250 000 digital water system data (data source: www.geodata.com)

as an ancillary model to construct the stream network model of the HTRW. We classified land use types into 27 categories. The main type of land use of

HTRW is forest (including orchard, 48.64%), dry land (24.38%), rice paddy (14.92%), urban land (vacant land, 7.78%) and unused land (uncultivated land, 1.85%) grassland (0.92%)

. Soil types were categorized into 26 types, the primary soil types in this area are brown soil (54.1%), meadow soil (29.7%)

and paddy soil (11.0%). The database of the underlying substrate was constructed based on the database of soil types using the land use data and soil data as underlying substrate parameters (Liu et al.,2015). The soil parameters were obtained from National earth system science data sharing infrastructure databaseNational earth system science data sharing infrastructure database was used to derive soil parameters. The watershed meteorological data  (precipitation, evaporation, and temperature)(daily precipitation and minimum and maximum air temperature data) used in the present study include precipitation data for 1990-2009 collected by 76 rainfall stations and air temperature data for1990-2009 collected by 12 city meteorological stations.

The missing meteorological information (rainfall, air temperature, relative humidity, mean wind velocity and solar radiation data) can be generated using the weather data generator simulation.The climate condition was then simulated using daily monitoring data from weather stations; weather data generator was used to supplement the missing records (missing rainfall, air temperature, relative humidity, mean wind velocity and solar radiation data) At least 3 sets monthly monitoring data . At least 3 data points per month for nitrate ($NO_3$), nitrite ($NO_2$), Ammonia ($NH_3$, $NH_4$), total nitrogen (TN), and total phosphorus (TP), were available in the time of 2006–2009were available for the period 2006–2009. Organic P and N were calculated by subtracting the sum of mineral components from the TP and TN, respectively (Neitsch,2005)

We got the information of crop type, farming system, sowing time, fertilization time, and social economics from investigation and statistics department in HTRW. Other information, including crop farming, tillage, social economics, and the amount and timing of fertilizer application, was based on a statistics yearbook, as well as on field investigations in HTRW. All the data were validated by the standard procedures used by the SWAT.

| Data type | Data scale | Data description | Data source | |
|-----------|-----------|------------------|-------------|---|
| DEM (Digital Elevation Model) | 1:250 000 | Elevation, overland and channel slopes and lengths | Institute of Geographical and Natural Resources Research; Chinese Academy of Sciences; National Geomatics Center of China | Input |
| Land use | 1:100 000 | Land use classifications | Institute of Geographical and Natural Resources Research; Chinese Academy of Sciences | Input |
| Soil properties | 1:1000 000 | Soil physical and chemical properties | Institute of Soil Science; Chinese Academy of Sciences | |
| Meteorological data | / | Precipitation, daily maximum and minimum air temperature, relative humidity and solar radiation | China Meteorological Administration; Liaoning province bureau of Meteorology | Input |
| water quantity and quality | / | / | Local hydrographical station and environmental monitoring station | Input |
| Social economic data | / | Population, livestock rearing, fertilizer application | Field investigation; Statistics yearbook | Input |

SWAT of HTRW

**Figure 2.** Data information in the HTRW.

| Data type | Scale | Data description | Source |
|-----------|-------|------------------|--------|
| Digital Elevation Model (DEM) | 1:250 000 | Elevation, overland and channel slopes and lengths | Institute of Geographical and Natural Resources Research; Chinese Academy of Sciences; National Geomatics Center of China |
| Land use | 1:100 000 | Land use classifications | Institute of Geographical and Natural Resources Research; Chinese Academy of Sciences |
| Soil properties | 1:1 000 000 | Soil physical and chemical properties | Institute of Soil Science; Chinese Academy of Sciences |
| Weather data | —— | Precipitation, daily maximum and minimum air temperature, relative humidity and solar radiation | China Meteorological Administration; Liaoning province bureau of Meteorology |
| water quantity and quality | —— | —— | Local hydrographical station and environmental monitoring station |
| Social economic data | —— | Population, livestock rearing, fertilizer application | Field investigation; Statistics yearbook |

The data information (type, scale, description, and source) of SWAT in HTRW are showed in Figure

2The details of the watershed features and the authority who issued the permission for information are listed in Table 1. We input the related meteorological, hydrological and soil data of SWAT got from China

Meteorological Administration, China Hydrology, and Environmental & Ecological Science Data Center for West China.We obtained the major GIS input files and the related physical data from Institute of

Geographical and Natural Resources Research, Institute of Soil Science and China Meteorological

Administration (Shen et al., 2013b). The China Hydrology, water resources & water quality monitoring department of HTRW provided the automatic & regular monitoring hydrological data

. The Liaoning province Water Resources Administrative Bureau granted permission for the modelling of the pollutant production response to different land utilization scenarios in the HTRW.

*2.2.3. Calibration and validation*

The data of monthly scale were used to achieve the simulation of SWAT

. We used the code open SWAT-CUP module to calibrate parameters of SWAT

in HTRW automatically (Abbaspour et al.,2007).

Sequential uncertainty fitting algorithm has higher calculation accuracy and simple application method, which was extensive used in the SWAT-CUP module (Wang et al.,2014; Yang et al.,2008)

. The $E_{NS}$ can effectively avoid the uncertainty of hydrological sequence (precipitation, water flow, and evaporation), which was used to evaluate the run-off flow change of hydrological station in HTRW

(Nash, 1970).

The model for the present study was calibrated and tested using artificial parameter modification and automatic calibration. First, the runoff was calibrated, followed by N, P and other nutrients. The runoff was calibrated and tested using real data from the Xingjiawopeng, and Tangmazai hydrological station (Figure

4). The simulated values of N and P were calibrated using monitoring data from Beikouqian, Dongling bridge, Xingjiawopeng, Xiaolinzi, and Tangmazhai hydrological station. Various hydrologic and water quality parameters were adjusted under their change interval to fit with the monitored/observed data during calibration and validation (Figure 3)Various hydrologic and water quality parameters were changed within their ranges to get the best fit with the observed data during calibration and validation (Table 2). ESCO,

GWQMN, and SURLAG were three key parameters in the process of calibration & validation of water flowESCO, GWQMN, and SURLAG were the three most sensitive parameters in the surface runoff simulation (Francos et al., 2003; Shen et al., 2010). For there was an explicit provision based on available water content in the soil profile, a change in the initial CN2 value would not greatly affect run off components. For nitrogen, the most sensitive parameters were NPERCO and SOL_NO3. For the phosphorus, SOL_LABP, PPERCO and PHOSKD were the most sensitive parameters. The initial concentration in the soil and the percolation coefficient were both identified to have a high degree of sensitivity for nutrients (Shen et al.,2014). The other sensitive parameters selected for calibration &

validation in HTRW were showed in Figure 3Based on the sensitivity analysis, the most sensitive parameters were screened for calibration and validation in HTRW. In the HTRW, Liaoning Province government began to monthly monitoring of pollutant since 2006 the local government began periodic monitoring of nutrients with approximately monthly sampling since 2006. The runoff, TN & TP loadings data used for calibration & validation were from 1992 to 2009, from 2006 to 2008, respectively.The parameter calibration and validation were conducted using data for runoff from 1992 to 2009, TN and TP

loadings from 2006 to 2008.

In the present study, the simulated effects were evaluated based on analysis and comparison using the runoff hydrograph, $Dv$ (relative deviation), $E_{NS}$ and $R^2$ (certainty coefficient)In the present study, the simulated effects were evaluated based on analysis and comparison using the runoff hydrograph, relative deviation ($Dv$), Nash-Sutcliffe efficiency coefficient ($E_{NS}$) and certainty coefficient ($R^2$). The runoff hydrograph and $Dv$ were frequently used to simulate the entire deviation of water quantity; $E_{NS}$ and $R^2$ were used to simulate the effects of the simulation (Yang et al.,2014). The $Dv$, $E_{NS}$ and $R^2$ are calculated as

$$D_v = [(M - W)/W] \times 100\% \qquad\qquad (1)$$

Here, $D_v$ was the relative deviation; $W$ was the observed mean value; and $M$ was the predicted mean value.

$$E_{NS} = 1 - [\sum_{i=1}^{n}(W_i - M_i)^2 / \sum_{i=1}^{n}(W_i - \overline{W})^2]$$ (2)

Here, $E_{NS}$ was the Nash-Sutcliffe efficiency coefficient; $W_i$ was the observed value at time $i$; $W_i$ was the simulated value at time $i$; and $\overline{W}$ was the observed mean value.

$$R^2 = \{[\sum_{i=1}^{n}(W_i - \overline{W})(M_i - \overline{M})] / [\sqrt{\sum_{i=1}^{n}(W_i - \overline{W})^2}\sqrt{\sum_{i=1}^{n}(M_i - \overline{M})^2}]\}^2$$ (3)

Here, $R^2$ was the certainty coefficient; Wᵢ was the observed data at $i^{th}$ period; $M_i$ was the simulated data at $i^{th}$ period; Wᵢ was the observed value at time i; Mᵢ was the simulated value at time i; $\overline{W}$ was the observed mean value; and $\overline{M}$ was the predicted mean value.

**Figure 3.** Parameters calibration of SWAT model in the HTRW

**Table 2.** Calibrated parameters of SWAT model in the HTRW. Based.

[revised manuscript text omitted]

**3.2.1. Sediment**

The sediment loading is the data basis to calculate the TN & TP loading, and which is affected by the type of land development and vegetation coverage (which was generally dominated by forest and farmland)The TN and TP loadings are closely linked with sediment loading, which is mainly affected by land use type (which was generally dominated by forest and farmland). Based on the simulation by the SWAT model, the annual output of sediment (silt) production in the watersheds of the Hunhe, Taizi and Daliao rivers was $22\times10^4$ t, $170\times10^4$ t and $30\times10^4$ t, respectively. The annual soil erosion modulus in the study area was 0.811 t/ha, and its spatial distribution is shown in Figure 46(a). The soil erosion (sediment) value varied widely in different regions, with the change interval from 0 to 1.824 t/ha.The amount of sediment (silt) yield was extremely different in sub-basin, with a range of 0-1.824 t/ha. Soil erosion in Daliao River watershed was very seriousThe soil erosion in the Daliao River watershed was the most severe (with up to 1.568 t/ha in some regions), followed by the Taizi River watershed (The amount was 1.223 t/ha in most regions) and Hunhe River watershed (Less than 0.19 t/ha in most regions). Yingkou and Dashiqiao has even topography, and incoming silt from the upper reaches is accumulated therein. The soil erosion modulus is therefore very high, which contributes greatly to the silt inputs to the HTRW (Tang et al.,2012). The soil erosion was affected by natural & human factors. The natural factors mainly included topography, underlying surface conditions and soil types, the human factors mainly consisted of vegetation coverage, precipitation type, land use, crop cultivation and cultivated land farming methods.The main factors affecting soil erosion included surface runoff, vegetation coverage rate and crop management, soil and water conservation practices and topographic factors (Ramos, 2006). Besides, the soil erosion modulus was relatively serious in agricultural land, particularly in dry land and paddy fields, based on a preliminary analysis. Moreover, mountainous area has great soil erosion (Hong et al.,2012). The HTRW had high forest coverage, which effectively prevented the soil erosionThe land use types were dominated by forest in HTRW, with relatively high vegetation coverage rate that prevented soil erosion. Daliao rivers had a large area of cultivated land, therefore, there was higher probability to cause soil erosionDaliao rivers was dominated by cultivated land, which was prone to soil erosion after rain scouring. Besides, the soil types are also the key influencing factors to cause soil erosion, therefore, the brown and paddy soils are prone to bring about the accumulation of sediment (Hong et al.,2012).The paddy soils, brown soils exported most sediment, which is much associated with the soil properties, indicating the strong penetration and low soil compaction (Hong et al.,2012). The soil types that are spatially distributed are a significant factor of high erosion.

[Figure]

a. The sediment.

[Figure]

b. The TP.

[Figure]

c. The TN.

**Fig 4.** NPS pollution loading distributions of HTRW under status quo scenario.

*3.2.2. Total Phosphorus (TP)*

With SWAT simulation results, the annual output of TP production in the watersheds of the Hunhe, Taizi and Daliao rivers was

8 993 t, 6 399 t and 3 315 t, respectively, the watershed loading output intensity was 7 kg/ha.

The TP loading had the same spatial distribution pattern with the sediment loading

.

The TP loading ranged from 0 to 259.83 kg/ha.

. Figure 6(b) showed the spatial variation of TP loading the HTRW

. The average annual water volume was affluent in Hunhe River, which prompted a large amount of P deposited in the downstream plain

. The changes in space of the TP loading was affected by topography

, precipitation, land use type, and silt losses. The TP loading output intensity of on the slope in the Daliao River watershed was higher than that in the Hunhe River watershed, and the Taizi River watershed was the lowest. Large amounts of fertilizer and pesticides have been applied to the farmland. Organophosphate pesticides accounted for 40%

of the total pesticides. Therefore, the farmland has high TP concentrations, which was the same findings with Wang(2012)Therefore, the farmland has high TP concentrations. The results are consistent with previous studies, proving that soil erosion is a significant contributor to NPS

pollution TP loading (Wang et al., 2012). The paddy fields and dry lands mainly distributed in

Hunhe River downstream, therefore, the P loading in these plain area is higher (Li et al.,

2010)The areas with high loading intensities of TP were concentrated in dry lands and paddy fields, with conventional tillage patterns and massive use of fertilizers, particularly in Hunhe

River downstream (Li et al., 2010). Correspondingly, the cities and counties with a large proportion of farmland, such as Dashiqiao, Panshan and Dawa city in the Daliao River watershed, as well as the city of Haicheng and Taian county in the Hunhe River watershed, have higher TP loading output intensity. The regions with a large proportion of developed land, such as the city center of Fushun, Shenyang in Hunhe River watershed, the municipal districts of Liaoyang city and Benxi city at the Taizi River watershed, which have lower TP loading output intensities. Based on the land use type, the tributaries with a higher proportion of farmland have the highest TP output intensities, whereas the tributaries with substantial vegetation cover as forested land have relatively lower TP output intensities. For the paddy soils, brown soils exported most sediment, the higher loading intensities of these soils are associated with the historic fertilizer application and the long residence times of nutrients in soil (Hong et al.,2012). The output intensity of TP is closely related to soil characteristics and attributes.The output intensity of TP is much associated with the soil properties.

**3.2.3. *TN*  ( *Total Nitrogen* )**

Upon simulation and calculation, the output of TN production in the watersheds of the Hunhe, Taizi and Daliao rivers was 24 264 t, 19 010 t and 10 048 t. The annual loading output intensity of TN in the watershed was 19 kg/ha. Figure 6(c) showed the spatial variation of TN loading the HTRW. The TN loading varied interval from 0.001 to 365.36 kg/ha. The TN loading had the same spatial characteristics with TP loading. The loading output intensity of TN in the Daliao River watershed was greater than that in the Hunhe River watershed, and the Taizi River watershed was the lowest. Large amounts of fertilizer were applied in the study area. Nitrate and organic N accounted a substantial portion of the fertilizer used in HTRW. Therefore, the loading output intensity of TN in the watershed was very high. The regions with a great proportion of farmland, such as the middle and lower reaches of the Hunhe River, the lower reaches of the Taizi River and the tributaries in the upper reaches of the Daliao River, have high output intensities of TN. The organic N contents in forested land was very low. Thus, the output intensity of TN in regions with high vegetation cover of forest, such as the mountainous area in the upper reaches of the Taizi and Hunhe rivers, was very low. The output loading intensity of TN in the municipal districts with high developed area was the lowest, such as the municipal districts of Fushun city and Shenyang city in the Hunhe River watershed, and the municipal districts of Benxi city, Liaoyang city and Shenyang city in the Taizi River watershed.

The loading intensity of TN and TP in the HTRW were characterized by its regional distribution. Although the counties of Qingyuan, Yibin and Benxi county, located in the upper reaches of the HTRW, had high output of water and silt, their loading intensities of pollution were not high. From the unit area perspective, the maximum loading intensities of TN and TP

were 365.36 and 259.83 kg/ha, respectively. The regions with high loading intensities of TN

and TP were mainly distributed in Taian, Haicheng, and Fushun city. The loading intensities of

TP and TN near the Dahuofang, Tanghe, Shenwo and Tanghe reservoirs were not high, ranging from 0.006-9.584 kg/ha, and 0.08-19.485 kg/ha, respectively. Based on the topography and soil type distribution, the gradient in the upper stream of HTRW was usually high<s>Based on the</s>

<s>topography and soil type distribution, the gradient in the upper reaches of the watershed was</s>

<s>usually high</s>. The soil type is predominately brown soil and salted paddy soil, both of which are easily eroded. The topography in the lower reaches is usually even, as in the cities of Anshan,

Haicheng, Yingkou and Panjin. The elevation is not high, and the soil type is usually predominately meadow soil and brown soil, both of which have a higher soil erosion rate, silt loss and loading intensity of pollutants. The regions with heavy loading intensities of TN and

TP included Xinmin county, located in the middle and lower reaches of the HTRW, the municipal district of Shenyang city, Liaozhong county, Dengta city, Liaoyang county, the municipal district of Anshan city, Haicheng city and a portion of Dashiqiao city. Based on the land development pattern in the Taizi River<s>Based on the land use type in the Taizi River</s>

<s>watershed</s>, dry fields and paddy fields were mainly distributed on the plain area of this watershed, which is therefore a core source of loading intensity. The spatial difference in the loading intensity between TN and TP were inconspicuous. soil types and land development status in the watershed, the upper stream of watershed have high vegetation coverage<s>Based on</s>

the topography, landform, soil types and land use conditions in the watershed, the upper reaches of the watershed have high vegetation coverage, less farmland and a low loading intensity of pollutants; the lower reaches of the watershed have more farmland, high rates of fertilizer application and a high soil erosion and pollution loading (Yin et al.,2011). To sum up, the spatial characteristics of TP loading was the result of comprehensive effect from precipitation/run off characteristics, soil properties, soil erosion and vegetation coverage. Consequently, the spatial distribution of sediment, TN and TP loadings is primarily related to the distribution of land and water, soil types and to the situation of local soil loss. Therefore, in order to effectively control TN loading and soil erosion in the HTRW, the BMPs, fallow measures of cultivated fields, watershed vegetation restoration and soil & water conservation in the upper stream, which were the most important measure that should be implemented Therefore, conscientious fertilization practices and eco-friendly tillage patterns should be implemented to control NPS pollution from farmlands, and the implementation of Best Management Practices (BMPs) for different NPS pollutant types should be considered in specific sites (Shen et al.,2014).

**3.3. NPS pollutant loading under EPSNPS pollution loading under environmental protection scenarios (EPS)**

The prevalence of farmland within a watershed has long been an important question, and strong evidence exists of a correlation between land development mode and water environment protect & rehabilitation at the basin scaleThe prevalence of farmland within a watershed has long been an important question, and strong evidence exists of a correlation between land use and water quality at the basin scale. Numerous studies have used land use data and stepwise regression analysis to explore relationships between land use and water quality parameters and ecological integrity on a regional scale, including sub-basins, river riparian buffer zones, and specific monitoring sites (Uriarte et al., 2011; Schiff, 2007; King et al., 2005)Numerous studies have used land use data and stepwise regression analysis to explore relationships between land use and water quality parameters and biotic integrity at multiple spatial scales, including sub- basins, river riparian buffer zones, and specific monitoring sites (Gove et al., 2001; Mehaffey et al., 2005; Uriarte et al., 2011; Li et al., 2009; Sliva, 2001; Schiff, 2007; King et al., 2005).

The riparian buffer zones could effectively reduce the concentration levels of $NO_3^-$ in water, which was 47% lower than the soil content (Venkatachalam et al.,2005)Study showed that the impact of riparian buffer zones was clearly observed in higher order streams where the observed $NO_3^-$ levels are 43.7% less than that of the upland (Venkatachalam et al.,2005). The dry farmland caused a higher NPS pollutant loading, followed by paddy, rural and urban area, forest land, and shrub landThe contribution rate of the NPS pollution loading coming from different land use types in descending order was dry farmland, paddy, rural and urban area, forest land, and shrub land. Under this developmental scenario, the area of farmland in the watershed was reduced; a modest area of farmland (29 500 ha, accounting for 2.74 % of the total farmland area) was converted to forestland (included shrub land, 14 753 ha), grassland (5

899 ha), wetland (8 848 ha); and NPS pollution from farmland decreased. The objective of water quality protection within the critical zoning of the watershed was realized. For the riparian buffers can be planted in various diverse vegetation, the N removal rate of 60m wide woody soil buffer zone was 16% and 38% higher than that of shrubbery and grassland, respectively (Aguiar et al.,2015). Considering different vegetation types used in the buffers,

60-m wide buffers composed of woody soils were more effective in N removal (99.9%) than areas with shrub (83.9%) or grass vegetation (61.6%) (Aguiar et al.,2015). Urban & rural areas were considered as the same type of land use in SWATRural and urban were treated as one

, about 1 kilometer within both banks of the tributaries of the Hunhe,

Taizi and Daliao rivers and 5 kilometers surrounding reservoirs were defined as buffer zones, including 1946 km$^2$ of farmland, urban land, and rural residential land, which accounts for 7.1 %

of the total area in the watershed. The woodland coverage rate was reduced by 1%, the loading intensity of sediment, TP and TN increased by 0.01~11.34, 0.15-2.83, and 0.40-14.00 kg/km$^2$, respectively~~For every 1% reduction in forest area, the loading intensity increased by 0.01~11.34~~

. The output of pollutant production under EPS was calculated by transforming the existing land use type.

Based on the parameter quantification results of SWAT, the TN and TP losses from farmland was effectively reduced after the modification of the land use structure. TN and TP respective range of change was from 0 to 365.357 kg/ha, and from 0 to 259.834 kg/ha

. The annual losses of TN and TP were reduced by 13 839 and 1 946 t/a, respectively. In comparison, the output of NPS pollutant production under the EPS was decreased by 21.9% compared with that under the status quo scenario, whereas the outputs of TP and TN were reduced by 10.4% and

25.9%, respectively. Under EPS, the average loading intensities of TN and TP were 14 and

6 kg/ha on a unit area basis, which were 14.3% and 26.3% less than the loading intensities under status quo scenario. The NPS pollution loading decline obviously in the EPS. The variation of

TP and TN pollution loading between status quo and EPS was shown in Table 4. The amount change indicated that riparian buffer and land development pattern change could effectively reduce the NPS pollutant loading in the HTRW.

**Table 4.** The variation of TP  & TN pollution loading between EPS and status quo scenario

| Watershed | Pollutant loading of EPS (kg/ha) | | Pollutant loading variation (kg/ha) | | Farmland variation (ha) | Forestland variation (ha) | Grassland variation (ha) | Wetland variation (ha) | Pollutant annual variation(t/a) | |
|---|---|---|---|---|---|---|---|---|---|---|
| | TP | TN | TP | TN | | | | | TP | TN |
| Hunhe River | 7 | 16 | -1 | -5 | -12460 | +6231 | +2492 | +3737 | -838 | -5743 |
| Taizi River | 4 | 10 | -1 | -4 | -14979 | +7491 | +2995 | +4493 | -776 | -5606 |
| Daliao River | 16 | 40 | -1 | -13 | -2061 | +1031 | +412 | +618 | -332 | -2490 |
| Total/Average | 6 | 14 | -1 | -5 | -29500 | +14753 | +5899 | +8848 | -1946 | -13839 |

"—" denotes a decrease compared to status quo scenario; "+" denotes an increase compared to status quo scenario.

**4. Conclusions**

The NPS pollution is prone to cause in dry farmland, paddy, rural & urban areas

. The SWAT model has been applied to study NPS in China by numerous research literature, they were mainly focuses on scenario simulation of NPS pollution and management in agricultural areas with rich hydrological and meteorological data. The basic monitoring data of HTRW were deficient, we selected the SWAT as the feasible method to access NPS pollutant loading in watershed level.

We applied certain practices based on

EPS to reduce the NPS pollutant loading in the Hunhe River, Taizi River and Daliao River watershed

. The status quo scenario and EPS were used to calculate the output of NPS pollutant production. Under the status quo scenario, the soil erosion modulus in the HTRW was 0.811 t/ha, and the soil erosion in the

Daliao River watershed was the most severe. The TP & TN annual loading in the HTRW was

19, and 7 kg/ha, respectively in the HTRW were 19, and 7 kg/ha, respectively. In the middle and lower stream of HTRW has a higher NPS pollutant loading, which included the urbanization and population density highly region of Shenyang, Anshan and LiaoyangThe region with a high NPS pollution loading is located in the middle and lower the HTRW, which included the urbanization and population density highly areas of Shenyang, Liaoyang and Anshan. Under the EPS, the TN and TP per unit area were 14, and 6 kg/ha, respectively. The output of NPS pollutant production, the loading intensities of TN and TP was reduced by 21.9%, 25.9% and 10.4% compared with the status quo scenario, respectively. In different regions of NPS pollutant loading in the HTRW changes greatly, and the pollutant loading intensity of different nutrients in the same region is slightly different. The NPS pollution occurring within different sub-basins and regions located in the watersheds varied greatly, and the loading intensities of different pollutant types in the given sub-basin were slightly different. Land eco-restoration and land development mode adjustment measures should be practiced to reduce NPS pollutant loading of cultivated landLand eco-restoration measures should be implemented to control agricultural NPS pollution from croplands. Therefore, SWAT simulation results provide a reference for the prevention of agricultural NPS pollution in agricultural watersheds.

In this study, the SWAT model can be used to simulate and calculate the source, and potential reduction of agricultural NPS pollutants based on different land use type. The reliability of SWAT evaluation results is decided by information completeness and the reasonable degree of parameter initializationThe SWAT simulation accuracy depends on the completeness of data and the reasonable degree of parameter initialization. In HTRW some data were missing, such as the rainfall intensity, and water pollution, et al. The data inaccuracy and local factors has a certain impact on SWAT model accession resultThe data inaccuracy and local factors would

. To determine the pollutant reduction under different land development patterns, and examine uncertainty of sensitivity parameters, SWAT model in China has wide range of potential application.

**Acknowledgements** The study was financially supported by the National Key Research and

Development Program of China (2016YFC0401408) and Project Based Personnel Exchange

Program with China Scholarship Council & German Academic Exchange Service of 2015. The author appreciates the experts & scholars of Helmholtz Centre for Environmental Research –

UFZ(Leipzig, Germany), as well as anonymous reviewers for their valuable comments and criticisms

.

---

## Short Comment (SC2) · 18 Jan 2018

After careful consideration, we feel that it has merit, but is not suitable for publication as it currently stands. Therefore, my decision is "Major Revision." General Content: There are numerous spelling mistakes and grammatical errors. This paper requires English editing and proof assistance. Authors should be careful using acronyms. If an acronym appears for the first time in the text, then it should be written in the full form. Ln 28: What is "pollutant production" mean? Did authors mean by pollutant generation? Ln 30: modulus? What does that mean, not sure why there is a mean and a modulus both? Again, could be something typo or just something else. Ln 30-31: output intensities? Why not only output? Or load? Or Concentration? Or flux? Overall this section is also difficult to read. Ln 31: intensities? Ln 95: please do not

refer readers to go and read literature from database. If there is literature pertinent to this paper, cite them; otherwise please do not direct readers. They can find any articles in Google scholar easily or other sites from libraries. Ln 83-85: there is no link between these two sentences. The last sentence may need to be a starting sentence for this paragraph instead. Ln 86: "The SWAT" of the present study"? Not sure what authors are trying to relate to? Ln 123: Inconsistency is writing proper names? For example, Atrazine is written with capital A, but not other fertilizers/pesticides such as acetachlor and butachlor? Ln 127: did the authors mean complete data? Statistics of what? Model Description: Ln 132: instead of "calculate", "predict" might be a better word. Ln 142: the threshold of 0% creates large number of HRUs to capture all heterogeneities. The reason behind using 0% threshold is not well justified. Model Inputs: Ln 188: 1:250 000, there is an additional space in 250000, it should be a comma. Calibration and Validation: Ln 216-218: why only NSE? Using NSE alone, as a performance indicator is not sufficient. It will not indicate any bias in model output. I am assuming that there are more metrics used. Ln 219: what is artificial parameter modification? Ln 221-225: What is the difference between "real data" and "monitoring data"? Aren't they both real? Ln 243-245: If the authors used SWAT-Cup for automatic calibration, then what is this manual calibration? Sensitivity analysis is included in SWAT-CUP. Did authors conduct a separate sensitivity analysis outside of SWAT-CUP? If so why? There is no need. Ln 244-250: Did the authors separate baseflow from total streamflow and calibrate runoff and baseflow separately? If so, the method needs to be clearly stated on how it was done? Manual or SWAT CUP? What is the point of using LOADEST program during the calibration. I understand that LOADEST could be used to calculate pollutant load, but what is unclear it the progam used during calibration. Ln283-299 In the preceding section, the authors mentioned that CN2 has no role. But it appears that runoff curve data was used for calibration. Please clarify. Results and Discussions: Similar observations could be made to TP and TN results and discussions. What are the common fertilizers used in the farmlands? Are there pastures and cattle feeding lots? What are

the initial soil nutrients content? Did the authors use this information for model parameterization? It is unclear from the methods and discussion. This paper could be improved if given time to rewrite everything. Paper could be published after revision.

Please also note the supplement to this comment:
https://www.hydrol-earth-syst-sci-discuss.net/hess-2017-755/hess-2017-755-SC2-supplement.pdf

---

## Author Comment (AC4) · 23 Jan 2018

Dear editors,

We would like to submit the enclosed manuscript entitled "*Reduction Assessment of Agricultural Non-Point Source Pollutant Loading*" (hess-2017-755), which we wish to be considered for publication in "*Hydrology and Earth System Sciences* (HESS)*"*.

No conflict of interest exits in the submission of this manuscript, and manuscript is approved by all authors for publication. In this work, we evaluated the manuscript is a part of our present research achievement, and which is a good paper. I hope this paper is suitable for "HESS". The main point our thesis wishes to address is to reflect on the practical application of and the solutions provided by the SWAT models in relation to China's sparse hydrological basin information; to provide point by point model constructions; an explanation of our process; an analysis of our results, and the expansion of the utilization of the SWAT model from an advanced and disciplined perspective. SWAT was used to assess the reduction of agricultural NPS pollutant. Buffer zone of land use type could reflect the natural environment. 21.9% pollutant reduction under the EPS.

We have tried our best to revise the manuscript to hope to meet with approval. The manuscript has been thoroughly checked again and revised as suggested with the help of an English teacher. It is believed that the revised paper will be readable and could meet the standard generally for publication.

Thank you very much for your consideration and help. Looking forward to hearing from you soon.

Thank you very much for your time and consideration.

Yours sincerely,

Dr. Yicheng Fu

E-mail: swfyc@126.com

+86-10-68781880 (office); +86-10-68572778 (fax)

Organization name: China Institute of Water Resources and Hydropower Research (IWHR)

Organization address: A-1 Fuxing Road, Haidian District, 100038 Beijing

23rd, Jan. 2018

**Additive list**

We have studied the valuable comments from you, the assistant editor and reviewers carefully, and tried our best to revise the manuscript. The point to point responds to the reviewer's comments are listed as following.

**Reviewer's Responses to Questions**

After careful consideration, we feel that it has merit, but is not suitable for publication as it currently stands. Therefore, my decision is "Major Revision."

(1)General Content: There are numerous spelling mistakes and grammatical errors. This paper requires English editing and proof assistance. Authors should be careful using acronyms. If an acronym appears for the first time in the text, then it should be written in the full form.

**Answer:** Thanks for your very thoughtful suggestion.

The manuscript has been thoroughly checked again and revised as suggested with the help of an English teacher (AJE, http://www.aje.com/). It is believed that the revised paper will be readable and could meet the standard generally for publication.

The EDITORIAL CERTIFICATE is followed.

**EDITORIAL CERTIFICATE**

This document certifies that the manuscript listed below was edited for proper English language, grammar, punctuation, spelling, and overall style by one or more of the highly qualified native English speaking editors at American Journal Experts.

**Manuscript title:**

An Estimation of Agricultural Surface-source Pollutant Production in Huntai River Watershed Based on the SWAT Model

**Authors:**

Y.C. Fu, J. Zhang, C.L. Zhang

**Date Issued:**

July 21, 2016

**Certificate Verification Key:**

B8D0-B6CF-FE85-D272-7AD3

[Figure]

This certificate may be verified at www.aje.com/certificate. This document certifies that the manuscript listed above was edited for proper English language, grammar, punctuation, spelling, and overall style by one or more of the highly qualified native English speaking editors at American Journal Experts. Neither the research content nor the authors' intentions were altered in any way during the editing process. Documents receiving this certification should be English-ready for publication; however, the author has the ability to accept or reject our suggestions and changes. To verify the final AJE edited version, please visit our verification page. If you have any questions or concerns about this edited document, please contact American Journal Experts at support@aje.com.

**An Estimation of Agricultural  Non-point  Source Pollut in the Huntai River Watershed Based on the SWAT Model**

Abstract: A SWAT model was constructed based on the rainfall runoff and land use types ; the migration-transformation process of agricultural  non-point source pollutants  were simulated and calculated, and the  export load and distribution traits of  non-point source pollutant were systematically analyzed based on the SWAT model  The SWAT model was calibrated and tested  using actual monitoring data as well as the physical property of the underlying substrate, hydrology, meteorology and pollutant sources in  Huntai Rver watershed. One kilometer within both banks of the trunk streams of the Huntai , Taizi  and Daliao rivers and 5 km  surrounding the reservoirs were defined as buffer zones. Existing land use types with the buffer zones were changed to restore the natural environment. The throughput of pollutant production under the regional  environmental protection priority  scenario was calculated based on the conventional development scenario. In the conventional development scenario, the annual mean modulus of soil erosion in the Huntai Rver watershed was 400 kg.hm⁻², and the output intensity of total N and P  were 19 and 7 kg.hm⁻², respectively. For the unit area, the maximal load intensity for total N and P  were 317 and 260 kg.hm⁻², respectively. Total N

批注 [Ed1]: Abbreviations and acronyms are often defined the first time they are used within the main text and then used throughout the remainder of the manuscript. Please consider adhering to this convention.

Besides, we have made serious changes to the expression of abbreviations in the whole paper, such as NPS (Non-point source), SWAT (Soil and Water Assessment Tool), TN (Total Nitrogen), TP (Total Phosphorus), HTRW (Huntai River Watershed), environmental protection scenario (EPS), DEM (Digital Elevation Model), and BMPs (Best Management Practices scenarios).

The revised contents could be found in the file of "paper revised version (clean)".

(2)Ln 28: What is "pollutant production" mean? Did authors mean by pollutant generation?

**Answer:** Thanks for your very thoughtful suggestion.

The mean of "pollutant production" is "pollutant generation". We have revised all the expressions in the paper. We have revised 23 places.

The revised contents could be found in the file of "paper revised version (clean)" & paper revised version (with track changes).

(3)Ln 30: modulus? What does that mean, not sure why there is a mean and a modulus both? Again, could be something typo or just something else.

**Answer:** Thanks for your very thoughtful suggestion.

The expression of "soil erosion modulus" is a proper noun.

In order to improve the readability of the article, reduce ambiguity, we have revised the sentence "the annual mean modulus of soil erosion in the HTRW was 811 kg/ha" to "the soil erosion modulus in the HTRW per year was 811 kg/ha".

In line 20, The systematically analysis contained three parts, which were (1) scenarios setting of SWAT; (2) modelling validation of SWAT in HTRW; (3) NPS

The revised section was as followed, Besides, the loadings and distribution traits of NPS pollutants were also systematically analyzed based on the model (scenarios setting, modelling validation, and pollutant loading calculation under status quo scenario & EPS).

In line 24, The "scenario settings" is the mean of "Land use types differences".

The revised contents could be found in the file of "paper revised version (clean)" & paper revised version (with track changes).

(4)Ln 30-31: output intensities? Why not only output? Or load? Or Concentration? Or flux? Overall this section is also difficult to read.

Ln 31: intensities?

**Answer:** Thanks for your very thoughtful suggestion.

The expression of "output intensities" is the mean of "output loading".

In order to improve the readability of the article, reduce ambiguity, we have revised the all the expressions in the whole paper, such as intensities.

The Ln28, we revised "output intensities" to "output loading".

The Ln29, we deleted the word of intensities.

The Ln35, we deleted the word of intensities.

The Ln83, we deleted the word of intensities.

The Ln57, we revised "output intensities" to "output loading".

The Ln65,66,67,68, we deleted the word of intensities.

Besides, we also revised some places such as intensity to loading, or deleted these words to make the article more fluent.

(5)Ln 83-85: there is no link between these two sentences. The last sentence may need to be a starting sentence for this paragraph instead.

**Answer:** Thanks for your very thoughtful suggestion.

We have changed the order of sentences. In order to make the expressions of the paragraph clearer, we have put the last sentence at the beginning of this paragraph.

(6)Ln 86: "The SWAT" of the present study"? Not sure what authors are trying to

relate to?

**Answer:** Thanks for your very thoughtful suggestion.

We have changed the expression of the sentence. In order to make the expressions of the content clearer, we have revised the sentence as the following,

"The SWAT Model was applied to quantify the output loading of TN & TP in HTRW under different land use types, assess the NPS pollutant loading reduction, and analyze the spatial distribution characteristics on the condition of land vegetation cover change.".

(7)Ln 95: please do not refer readers to go and read literature from database. If there is literature pertinent to this paper, cite them; otherwise please do not direct readers. They can find any articles in Google scholar easily or other sites from libraries.

**Answer:** Thanks for your very thoughtful suggestion.

We have We have shortened the length of Section 2.1. We only provided the necessary information of study area. The contents could be found as following,

The HTRW (40°27′~42°19′N, 121°57′~125°20′E) is in Liaoning province (Northeast China), and the watershed area is 2.73×104 km2, which takes about 1/5 of the area of Liaoning province (Fig 1). The HTRW is a tributary of Liaohe River Basin (The Liaohe River Basin is one of China's larger water systems) and is consist of Hunhe River, Taizi River, and Daliao River. The Hunhe River, Taizi River, and Daliao River watershed is HTRW's sub-watershed. The HTRW has varied topography, low mountain is located in eastern part, and the other parts are alluvial plain. The elevation of northeast region is high. Loamy soils are mainly distributed in alluvial plain, and the average grade of lower HTRW is about 7%. HTRW area includes the cities of Fushun, Shenyang, Benxi, Liaoyang, Anshan, and Yingkou, most of Panjin city, some portions of Tieling city and a minor portion of Dandong city. The stream flow and nutrient were validated based on the five monitoring stations, Beikouqian, Dongling Bridge and Xingjiawopeng are located in Hunhe River, Xialinzi and Tangmazhai are in Taizi Rive. HTRW has temperate continental climate, the average annual temperature is 7°C, and precipitation is 748 mm.

Besides, we have put the basic information of SWAT model, and the operation & application procedure show in the form of pictures (Figure 2, Figure 3), to reduce the length of the article, and increase the readability of the article. We also cited the relevant literature to add the scientific and readability of the paper, such as Ln 137, L138, L140, and L145, et. al.

(8)Ln 123: Inconsistency is writing proper names? For example, Atrazine is written with capital A, but not other fertilizers/pesticides such as acetachlor and butachlor?

**Answer:** Thanks for your very thoughtful suggestion.

The words of "diammonium phosphate, potassium phosphate, atrazine and acetochlor" were proper names. We have revised the form of these words in the form of special nouns. Such as

Ln 125-127, The heavy use of chemical fertilizers was mainly urea, DAP (diammonium phosphate) & a small amount of N-P-K (Nitrogen-Phosphorus-Potassium mixed fertilizer).

Ln 127-130, Atrazine & Acetochlor were mainly used on dry farmland, and Butachlor was mainly used in paddy fields. Based on the statistical data for 2006-2012, the quantity of fertilizers and Pesticides (such as Methamidophos & Plifenate) applied in the watershed fluctuated annually.

(9)Ln 127: did the authors mean complete data? Statistics of what?

**Answer:** Thanks for your very thoughtful suggestion.

We revised the ambiguous expression. The revised sentence is as follows,

"and therefore statistical department statistics (http://www.ln.stats.gov.cn/tjsj/tjgb/, http://www.stats.gov.cn/) are not available.". We obtained these data & information which would normally be inaccessible through the on-site investigation, inquiry visits, case studies, example analysis, expert consultation and material research method.

(10)Model Description:

Ln 132: instead of "calculate", "predict" might be a better word.

**Answer:** Thanks for your very thoughtful suggestion.

We have revised the word "calculate" to "predict".

(11)Ln 142: the threshold of 0% creates large number of HRUs to capture all heterogeneities. The reason behind using 0% threshold is not well justified.

**Answer:** Thanks for your very thoughtful suggestion.

We have added 0% as the basis for generating HRUs. The added contents were as follows,

HRU is the minimum unit to predict pollutant output loading, which is automatically generated by the superimposition of land use & soil types within the sub-river basin. Because lots of HRUs were automatically generated based on different combinations,we selected 0% underlying surface data as the initial value to generate HUR that is consistent with the distribution characteristics of HTRW water system.

(12)Model Inputs:

(12)Ln 188: 1:250 000, there is an additional space in 250000, it should be a comma.

**Answer:** Thanks for your very thoughtful suggestion.

We have revised the space to a comma in 250000.

We also revised the other data in the whole paper.

(13)Calibration and Validation:

Ln 216-218: why only NSE? Using NSE alone, as a performance indicator is not sufficient. It will not indicate any bias in model output. I am assuming that there are more metrics used.

**Answer:** Thanks for your very thoughtful suggestion.

We have added the relative contents as follows,

We used the open SWAT-CUP software to adjust parameters, then the SUFI-2 algorithm was selected to determine the optimal values of the parameters based on iterative computations. Finally, we manually input the optimal parameters to SWAT model for hydrology series simulation. The $E_{NS}$ (Nash-Sutcliffe efficiency coefficient), $Dv$ (relative deviation), and $R^2$ (certainty coefficient) can effectively avoid the uncertainty of hydrological sequence (precipitation, water flow, and evaporation),

which was used to evaluate the run-off flow change of hydrological station in HTRW.

(14)Ln 219: what is artificial parameter modification?

**Answer:** Thanks for your very thoughtful suggestion.

We revised the ambiguous expression. The revised sentence is as follows,

The SWAT for the present study was calibrated and tested using the coupled method of manual & auto-calibration. The uncertainty analysis was carried out by using SWAT-CUP program.

(15)Ln 221-225: What is the difference between "real data" and "monitoring data"? Aren't they both real?

**Answer:** Thanks for your very thoughtful suggestion.

In the paper, the "real data" is the "monitoring data". We have revised the ambiguous expression. The revised sentence is as follows,

The runoff was calibrated and tested using monitoring data from the Xingjiawopeng, and Tangmazai hydrological station (Figure 4).

(16)Ln 243-245: If the authors used SWAT-Cup for automatic calibration, then what is this manual calibration? Sensitivity analysis is included in SWAT-CUP. Did authors conduct a separate sensitivity analysis outside of SWAT-CUP? If so why? There is no need.

**Answer:** Thanks for your very thoughtful suggestion.

In the paper, we used the coupled method of manual & auto-calibration to analyze the parameters sensitivity. The revised sentence is as follows,

Sensitivity analysis of the parameters is an effective means to reduce the uncertainty of the hydrological model and increase effectiveness of SWAT model. The sensitivity evaluation indicators are different among SWAT and SWAT-CUP. The "T-test (Student's t test)" used by SWAT-CUP is part-sensitive. We draw on manual calibration analysis to make necessary adjustments to the SWAT-CUP sensitivity analysis. To improve the accuracy of model calibration & verification results, we used SWAT-CUP and SUFI-2 algorithm to analyze the parameters sensitivity.

(17)Ln 244-250: Did the authors separate baseflow from total streamflow and calibrate runoff and baseflow separately? If so, the method needs to be clearly stated on how it was done? Manual or SWAT CUP? What is the point of using LOADEST program during the calibration. I understand that LOADEST could be used to calculate pollutant load, but what is unclear it the program used during calibration.

**Answer:** Thanks for your very thoughtful suggestion.

In the paper, we have revised the ambiguous expression. The revised sentence is as follows,

(1) Direct runoff is surface runoff resulting from rainfall, which includes surface and return flows. Baseflow is part of groundwater recharge to river runoff. It is impossible to measure or directly divide the base flow from the total runoff. Most of the base flow and direct runoff segmentation are performed by mathematical methods. We used Digital-Filter-Equation to divide the base flow (Lyne & Hollick, 1979).

$$\begin{cases} q_t = \beta.q_{t-1} + \alpha(1+\beta)(Q_t - Q_{t-1}) \\ b_t = Q_t - q_t \end{cases} \tag{1}$$

Here, $q_t$ is the surface runoff at time $t$; $Q_t$ is the total runoff at time $t$; $b_t$ is the base flow at time $t$; $\alpha$ & $\beta$ are filter parameter.

According to the following steps. (1) The first filter made the second record points as a starting point forward calculate one by one. We calculated $q_t$ according to equation (1), and $b_t \geqslant 0$, $q_t \geqslant 0$, $Q_t \geqslant b_t$, $Q_t \geqslant q_t$. If $q_t < 0$ at time $t$, we assigned $q_t = 0$, $Q_t = b_t$; If $b_t < 0$ at time $t$, we assigned $b_t = 0$; If $b_t > Q_t$ at time $t$, we assigned $b_t = q_t$. (2) The second filter made the penultimate record points as a starting point backward calculate based on the calculation result of first filter. (3) The calculation of third filter following the positive operation. According to the above calculation rules, we divided the base flow until getting the smooth base flow process line. Digital filtering is an objective & effective method of base-stream segmentation, we assigned $\alpha$ =0.5, $\beta$ =0.925 in HTRW (Arnold & Allen,1999).

(2) We used the Auto-Calibration & Uncertainty module (SWAT-CUP) of SWAT to automatically calibrate 10 sensitive parameters, then we applied the Manual

calibration helper of SWAT to make small & targeted adjustments to the calibration results to improve the simulation accuracy based on auto-calibration results. The contents could be found in Ln 247-250 (paper revised version (clean)).

The reference as following,

Arnold, J., Allen, P.: Automated Methods for Estimating Baseflow and Ground Water Recharge from Streamflow Records. Journal of the American Water Resources Association, 35, 411-424, 1999.

Lyne, V.D., Hollick, M.: Stochastic Time Variable Rainfall-Runoff Modeling: Proceedings of Hydrology and Water Resources Symposium. Perth, Australia: National Committee on Hydrology and Water Resources of the Institution of Engineers,1979.

(3) We have revised the expression in Ln 249.

During calibration, we used $R^2$ & correlation coefficient of residual sequence (*SCR*) to eliminate the uncertainties caused by the differences in sampling & testing methods of water quality (Yang et al.,2014).

(18) Ln283-299 In the preceding section, the authors mentioned that CN2 has no role. But it appears that runoff curve data was used for calibration. Please clarify.

**Answer:** Thanks for your very thoughtful suggestion.

In the paper, we have added the related contents as follows,

CN2 is a comprehensive parameter that reflects the characteristics of watershed before rainfall. It is mainly affected by the hydrology & soil types, land use types, pre-soil moisture and tillage management measures. CN2 directly affects the surface runoff. The larger the CN2 value, the larger the runoff yield. With the same type of land use, the greater the permeability, the smaller the CN2 value. With the same type of land use, the lower vegetation coverage & rainfall interception ability, the greater the CN2 value. Different regions have different CN2 values, the moist area is the highest, the range of 60~96, while other regions vary greatly. With the same soil types, CN2 value of cultivated land was the highest, followed by grassland and woodland was the lowest.

(19)Results and Discussions:

Similar observations could be made to TP and TN results and discussions. What are the common fertilizers used in the farmlands? Are there pastures and cattle feeding lots? What are the initial soil nutrients content? Did the authors use this information for model parameterization? It is unclear from the methods and discussion.

**Answer:** Thanks for your very thoughtful suggestion.

In the paper, we have added the related contents as follows,

(1) Considering land pattern, rainfall and source of pollutants, the HTRW faces a high risk of pollution from agriculture. Heavy use of fertilizers and soil erosion in the upper of HTRW has led to serious NPS pollution in HTRW.

Fertilization in the HTRW is predominantly with nitrogen, followed by phosphorous and potassium. The heavy use of chemical fertilizers was mainly urea, DAP (diammonium phosphate) & a small amount of N-P-K (Nitrogen-Phosphorus-Potassium mixed fertilizer). Atrazine & Acetochlor were mainly used on dry farmland, and Butachlor was mainly used in paddy fields. Based on the statistical data for 2006-2012, the quantity of fertilizers and Pesticides (such as Methamidophos & Plifenate) applied in the watershed fluctuated annually.

These contents could be found in the Ln 121-129 in the revised version (clean).

(2) Brown soil is widely distributed in the HTRW. We supplied the characteristics of N & P loss under different land use types and fertilization, as shown in Table 2. The thickness of brown soil was 30-50cm in HTRW. The content of organic, TN & TP decreased significantly with the increment of soil depth. Nutrients were mainly found in soils of 0-30cm depth, where TN & TP reserves reached more than 50% of the total reserves in the soil.

**Table 2.** The loss characteristics of N & P under different land use types & fertilization

| Land use type | Soil thickness (cm) | Organic matter content (g/kg) | Unit weight of soil (g/cm$^3$) | Soil particle composition | | | TN (g/kg) | TP (g/kg) |
|---|---|---|---|---|---|---|---|---|
| | | | | Cosmid ∅≤0.002 | Powder 0.002<∅≤0.005 | Sand 0.005<∅≤2 | | |
| cultivated | 0-5 | 24.58 | 1.42 | 21.05 | 57.35 | 21.6 | 0.96 | 0.47 |
| field | 5-30 | 18.45 | 1.48 | 24.71 | 24.71 | 18.84 | 0.88 | 0.38 |
| Grassland | 0-5 | 27.6 | 1.18 | 15.97 | 15.97 | 14.58 | 1.25 | 0.58 |

| | 5-30 | 21.75 | 1.25 | 20.36 | 20.36 | 21.5 | 1.02 | 0.42 |

Reference:Hao, L.P.: Characteristics of nitrogen and phosphorus losses of rainfall runoff in Liaoning Hunhe Basin, Shenyang Agricultural University, 2012.

The large-scale use of fertilizers (DAP, N:46.4% & N-P-K, N:15%; P2O5:15%; K2O:15%) & livestock and poultry excrement (N:0.5-0.6%; P:0.45-0.6%; K:0.35-0.5%) were the important sources of agricultural NPS pollution. In HTRW, the numbers of pastures and cattle was little, and the excretions of cattle were collected and processed by the farmer.

The excessive or unreasonable application of fertilizers, and the fertilizer utilization rate was not high (the utilization rate of nitrogen is 30% to 60%, and phosphorus is 2% to 25%), resulting in a large number of fertilizer loss. The nutrient content (mainly from agricultural production activities) of soil in HTRW was 1.21t/ha.

The information of initial soil nutrients content & fertilizers was used for model parameterization.

*We tried our best to improve the manuscript and made some changes in the manuscript. These changes will not influence the content and framework of the paper. And here we did not list the changes but marked in red in revised paper (Revision, changes marked).*

*We appreciate for Editors/Reviewers' warm work earnestly, and hope that the correction will meet with approval.*

*Once again, thank you very much for your comments and suggestions.*

---

## Short Comment (SC3) · 21 Feb 2018

Following an explicit e-mail request by the Authors to withdraw the manuscript, the discussion process can be stopped.

---

## Author Comment (AC5) · 23 Feb 2018

Dear editor, Please excuse me for taking some of your time. Thank you very much for your kind remind and helps. But now we feel that we have not yet studied our work completely and some new great results are discovered. So, after carefully thinking, we are going to rearrange this manuscript and try to give more precise model. Thus, we are sorry that we have to withdraw the submission of our paper "Reduction Assessment of Agricultural Non-Point Source Pollutant Loading" (No. hess-2017-755). The reason is that, first, if we revise the paper according to the reviewers' comments point by point, we will pay much time to make tests and then plot the data in the revised paper; second, we feel that we have not yet studied our work completely and have some difficulties to answer some questions of the reviewers' comments and add their

answers or solutions in the revised paper quickly. Of course, our work in this paper is complex and difficult because it is related with the simulation, assessment, calculation, management and some couplings among them. I am very sorry that we had to make this decision and continue our investigation of this work further. I appreciate the reviewers very much for their careful work and helpful suggestions. Thank you very much for your consideration and help. Of course, after repreparation, in your journal you will find this lovely manuscript, and we hope it comes soon. We sincerely say sorry for all the staffs involved this manuscript, including editor, reviewers, etc. because of our action. Looking forward to hearing from you soon. Thank you very much for your time and consideration.

Yours sincerely, Dr. Yicheng Fu

E-mail: swfyc@126.com +86-10-68781880 (office); +86-10-68572778 (fax) Organization name: China Institute of Water Resources and Hydropower Research (IWHR) Organization address: A-1 Fuxing Road, Haidian District, 100038 Beijing

23rd, Feb. 2018